# Structural connectome architecture shapes the maturation of cortical morphology from childhood to adolescence

Xinyuan Liang [1,2,3], Lianglong Sun[1,2,3], Xuhong Liao [4], Tianyuan Lei [1,2,3], Mingrui Xia [1,2,3], Dingna Duan[1,2,3], Zilong Zeng[1,2,3], Qiongling Li [1,2,3], Zhilei Xu[1,2,3], Weiwei Men[5,6], Yanpei Wang[1], Shuping Tan[7], Jia-Hong Gao [5,6,8], Shaozheng Qin [1,2,3,9], Sha Tao [1], Qi Dong[1], Tengda Zhao [1,2,3] ✉ & Yong He [1,2,3,9] ✉

Cortical thinning is an important hallmark of the maturation of brain morphology during childhood and adolescence. However, the connectome-based wiring mechanism that underlies cortical maturation remains unclear. Here, we show cortical thinning patterns primarily located in the lateral frontal and parietal heteromodal nodes during childhood and adolescence, which are structurally constrained by white matter network architecture and are particularly represented using a network-based diffusion model. Furthermore, connectome-based constraints are regionally heterogeneous, with the largest constraints residing in frontoparietal nodes, and are associated with gene expression signatures of microstructural neurodevelopmental events. These results are highly reproducible in another independent dataset. These findings advance our understanding of network-level mechanisms and the associated genetic basis that underlies the maturational process of cortical morphology during childhood and adolescence.

The transition period from childhood to adolescence is characterized by prominent reorganization of cortical morphology[1,2], which provides critical support for cognitive growth[3,4]. With progress in modern in vivo structural brain imaging, researchers have documented widespread spatial refinements of cortical morphology during childhood and adolescence[5,6]. A typical cortical maturation sequence is marked by hierarchical cortical thinning from the primary cortex to the association cortex[1,7,8] and is thought to be mediated by cellular mechanisms, genetic regulation, and biomechanical factors[9,10]. From a multifaceted developmental perspective,

anatomical refinements within neuronal layers at local gyri and sulci, such as synaptic pruning and myelination[11,12], as well as the tension exerted by white matter (WM) fibers[13,14], could collectively contribute to cortical maturation. Specifically, WM pathways serve as a structural scaffold for interregional communication, playing a crucial role in supporting the intricate interplay among these factors. Understanding how the brain WM scaffold facilitates anatomical refinements can provide new insights into maturational principles of cortical morphology. In the present study, we present a mechanistic approach to model how the maturational pattern of cortical

[1]State Key Laboratory of Cognitive Neuroscience and Learning, Beijing Normal University, Beijing 100875, China. [2]Beijing Key Laboratory of Brain Imaging and Connectomics, Beijing Normal University, Beijing 100875, China. [3]IDG/McGovern Institute for Brain Research, Beijing Normal University, Beijing 100875, China. [4]School of Systems Science, Beijing Normal University, Beijing 100875, China. [5]Center for MRI Research, Academy for Advanced Interdisciplinary Studies, Peking University, Beijing 100871, China. [6]Beijing City Key Laboratory for Medical Physics and Engineering, Institute of Heavy Ion Physics, School of Physics, Peking University, Beijing 100871, China. [7]Beijing Huilongguan Hospital, Peking University Huilongguan Clinical Medical School, Beijing 100096, China. [8]IDG/McGovern Institute for Brain Research, Peking University, Beijing 100871, China. [9]Chinese Institute for Brain Research, Beijing 102206, China. ✉e-mail: tengdazhao@bnu.edu.cn; yong.he@bnu.edu.cn

morphology is shaped by WM connectome architecture from childhood to adolescence.

At the microscale level, numerous histological studies have suggested that the brain's WM pathways are involved in the developmental process of cortical gray matter. During neural circuit formation, axons express guidance receptors to integrate attractive and repulsive environmental information for navigation to their target neurons[15,16]. After axons arrive, synaptic maintenance and plasticity rely on active axonal transport through axonal cytoskeletons, which offers essential delivery of neurotrophic factors, energy requirements, and synthesized or degraded proteins for long-distance cortical neurons[17–19]. Such early-established neuronal pathways could lead to preferences in attracting or removing new links during the formation of cortical hubs[20]. Physical simulation studies have suggested that there is a tension-induced relationship between fiber growth and cortical fold morphology[13,14]. At a macroscale level, several prior studies using structural and diffusion magnetic resonance imaging (MRI) have also shown that decreases in focal cortical thickness (CT) are associated with increased microstructural anisotropy and decreased mean diffusivity in adjacent WM[21–24] and that homologous cortical regions, which are rich in WM fibers, exhibit stronger maturational couplings of CT than non-homologous regions[23,25]. Notably, all these previous studies are limited to local cortical regions or specific fiber tracts. The human brain is a highly interactive network in nature in which connections promote information exchange between brain regions, raising the possibility that the maturation of focal cortical morphology is shaped by the overall architecture of the WM connectome. However, whether and how the maturation pattern of cortical morphology from childhood to adolescence is constrained by physical network structures, and specifically, whether this constraint works following a network-based diffusion model, remains largely unknown. We anticipate that models of regional cortical maturation would yield mechanistic insights into the network structure that governs the coordinated development of cortical morphology among regions.

If the connectome structure shapes regional cortical maturation, it is necessary to further clarify whether this constraint is associated with genetic factors. Converging evidence indicates that genetic modulations may exist on the potential constraint of WM maturation on cortical morphology. Studies on the rodent nervous system[26,27] have shown that the wiring diagram of brains is tracked by genes that are involved in axon guidance and neuronal development processes. Such genetic cues are also related to molecules responsible for cytoskeletal rearrangements that induce cortical refinement processes, including synaptic pruning and neuron cell death[16]. In humans, recent emerging transcriptome imaging analyses pave a new way to link brain macroscale structural maturation to microscale biological processes by seeking linkages between MRI-based brain measurements and genetic samples of postmortem brains. Such studies have shown that cortical thinning during development is related to genes involved in the structure and function of synapses, dendrites, and myelin[28,29]. These precisely programmed microstructural alterations constitute major neurodevelopmental events that promote the establishment of more mature brain architecture and anatomical connectivity from childhood to adolescence[30,31]. Therefore, we further hypothesize that the constraint between the maturation of cortical morphology and WM network structure is associated with gene expression profiles that are involved in neurodevelopment.

To fill these gaps, in the present study, we integrated neuroimaging, connectome, and transcriptome analyses as well as computational modeling to investigate the network-level mechanisms underlying regional changes in cortical morphology during childhood and adolescence and to further explore their potential genetic underpinnings. Specifically, we tested three hypotheses: (i) that the maturation of CT in brain nodes is associated with that of structurally connected neighbors, (ii) that the network-level diffusion model, which represents the direct and high-order information exchange preferences among neighbors, captures the principle of connectome constraint on the maturation of CT, and (iii) that the connectome constraints on cortical maturation are linked with gene expression levels of neurodevelopment processes.

## Results

### Data samples

To investigate the relationship between cortical morphology maturation and the WM connectome from childhood to adolescence, we leveraged structural and diffusion MRI data from a longitudinal MRI dataset ("discovery dataset") containing 521 brain scans from 314 participants (aged 6–14 years) in the Children School Functions and Brain Development Project in China (Beijing Cohort) (Fig. 1a). Among the participants, 158 underwent a single scan, 105 underwent two scans separated by an average interval of 1.16 years, and 51 underwent three scans separated by an average interval of 0.99 years.

Currently, it is not clear which statistical models best capture cortical development over time. Therefore, in the present study, we employed three distinct statistical models to assess nodal CT maturation from childhood to adolescence, which include group-wise comparisons between children and adolescents with a mixed linear model[32] (Statistical Model I), generalized additive model (GAM) analysis[33] including age as a continuous variable (Statistical Model II), and individual-level longitudinal analysis using repeated brain imaging scans (Statistical Model III). Our comprehensive analyses aim to yield robust conclusions about the maturational pattern of cortical morphology from childhood to adolescence and how the structural connectome architecture shapes cortical maturation. In Statistical Model I, we employed a group-wise comparison to examine the critical transition from childhood to adolescence. In this analysis, we divided all participants into the child group (218 participants, 299 scans, 6.08–9.98 y) and the adolescent group (162 participants, 222 scans, 10.00–13.99 y) using the age of 10 years as a cutoff, according to the criteria from a previous public health investigation[34] and the World Health Organization (WHO)[35]. This method has an advantage in that it is less sensitive to the age distribution of individuals and does not require prior models for fitting age-related growth curves of CT; however, it loses some power to capture fine-grained, age-related change trajectories of CT maturation[5,36]. In Statistical Model II, we employed a GAM analysis that treated age as a continuous variable to chart the fine-grained cortical maturation patterns. In this way, we obtained age-related change curves of cortical maturation and investigated how these morphological refinements were constrained by the WM network architecture across different ages. However, this method is sensitive to the sample sizes for each age. In Statistical Model III, we leveraged longitudinal structural MRI data from participants who underwent two or three repeated scans to examine the individual differences in cortical maturation. This individual-based analysis focused on capturing pure longitudinal changes within individuals, minimizing intersubject confounds such as lifestyle and genetic factors[37–39]. To assess the reproducibility of our results, we included an independent dataset ("replication dataset") that contains cross-sectional structural and diffusion MRI data from 301 typically developing participants selected from the Lifespan Human Connectome Project in Development (HCP-D)[40]. Statistical Models I and II were applied to the cross-sectional replication dataset. Details on the demographic information of all participants, data acquisition, and data analysis are provided in the Methods section and Supplementary Sections 1.1–1.2.

### Typical spatial refinement of brain CT from childhood to adolescence

For each individual, we first parcellated the brain cortex into 1000 nodes of interest with approximately equal size (219 and 448 node

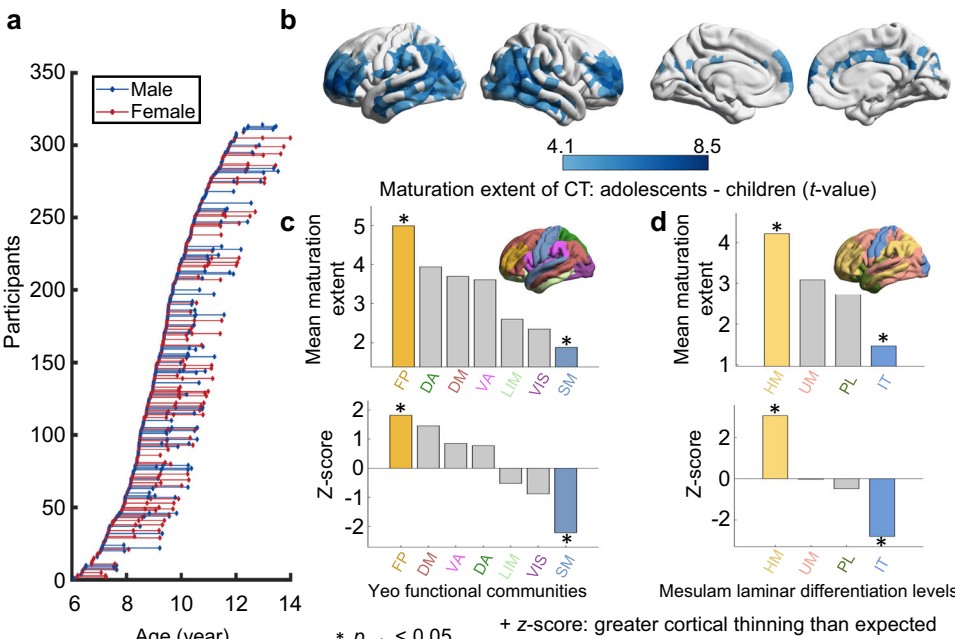

**Fig. 1 | Distribution of age at scan among all participants and typical CT maturation from childhood to adolescence. a** Participant age at MRI scan(s). Each dot represents a single scan of a participant and the longitudinal scans from the same participant are connected by lines. **b** Brain map of regional CT differences between the child and adolescent groups. Group differences were used to represent the extent of CT maturation in a mixed linear analysis including sex as a covariate and the individual-specific intercept as a random effect (Statistical Model I, Bonferroni corrected at $p = 5 \times 10^{-5}$, two-sided). A greater positive $t$ value (darker blue) denotes more pronounced cortical thinning with development. **c, d** The mean CT maturation extent (estimated by $t$ value) within each brain community was defined by Yeo et al., [44] and the laminar differentiation level was defined by

Mesulam et al., [47]. Spin tests[45,46] were performed by spherical projection and rotation class positions 1000 times for correcting spatial autocorrelations, and the class-specific mean $t$ values were expressed as $z$ scores relative to this null model. A positive $z$ score indicates greater cortical thinning than expected by chance. Asterisks denote significant level at $p_{spin} < 0.05$ ($p_{spin}$(FP) = 0.029, $p_{spin}$(SM) = 0.004, $p_{spin}$(HM) < 0.001, and $p_{spin}$(IT) = 0.001, one-sided). VIS, visual; SM, somatomotor; LIM, limbic; DA, dorsal attention; VA, ventral attention; FP, frontoparietal; DM, default mode; IT, idiotypic; PL, paralimbic; UM, unimodal and HM, heteromodal. Values of the brain map were visualized using BrainNet Viewer (1.7)[140].

parcellations as a validation[41]) according to the modified Desikan-Kiliany atlas[42,43]. Then, we computed the average CT for each brain node based on structural MR images in the FreeSurfer v6.0 image analysis suite (https://surfer.nmr.mgh.harvard.edu/) (for details, see the Methods). To delineate the spatial maturation map of brain morphology, we estimated the statistical differences in regional CT between the child and adolescent groups to represent the CT maturation extent by a mixed linear analysis[32] with sex included as a covariate (Statistical Model I). Brain nodes showed significant cortical thinning, mainly concentrated in dorsolateral prefrontal regions, lateral temporal and lateral parietal regions (Fig. 1b, $t$ values > 4.10, $P < 0.05$, Bonferroni corrected). To test whether this maturation pattern is anchored to specific brain systems, we classified all cortical nodes into seven well-validated brain communities[44] and performed a spherical projection null test ("spin test") to correct for spatial autocorrelations by permuting communities positions 1000 times[45,46]. The class-specific mean $t$-values were expressed as $z$ scores relative to this null model. We found that all brain systems showed decreased CT with development on average. The frontoparietal (FP) system and default mode (DM) system showed higher cortical thinning than expected by chance (FP: $p_{spin} = 0.029$; DM: $p_{spin} = 0.068$, Fig. 1c). The somatomotor (SM) system displayed lower cortical thinning than expected by chance ($p_{spin} = 0.004$). We also repeated this analysis by classifying cortical nodes into four laminar differentiation levels[47]. We found that heteromodal areas displayed cortical thinning ($p_{spin} < 0.001$), while idiotypic areas showed lower cortical thinning than expected by chance ($p_{spin} = 0.001$, Fig. 1d). Consistent results were found at the other two parcellation resolutions (Supplementary Figs. S1, S2). These results are largely compatible with previous studies[1,8], demonstrating

that CT exhibits the most pronounced thinning in high-order association areas and is relatively preserved in primary areas from childhood to adolescence.

## Spatial maturation of CT links with direct WM connections

Next, we tested whether the regional maturation of CT was associated with the WM network architecture. To this end, we first reconstructed individual structural connectomes with 1000 nodes (219 and 448 nodes were used for validation) based on diffusion MR images of the child group by performing deterministic tractography between cortical regions[48,49]. We then generated a binary, group-level connectome by using a consensus approach that preserved the connection length distributions of individual networks[50] (Fig. 2a, Supplementary Section 2.1).

Next, we estimated the across-node relationship of the CT maturation extent ($t$-value between the child and adolescent group, derived from Statistical Model I) between a node and its directly connected neighbor nodes in the backbone (Fig. 2b). We found a significant spatial correlation between the nodal CT maturation extent and the mean of its directly connected neighbors (Fig. 2c, adjusted $r = 0.74$, $P = 5.56 \times 10^{-176}$). Next, we tested the significance of this spatial correlation against two baseline null models. The first model evaluated whether the observed correlation was determined by the wiring topology rather than the basic spatial embedding of the WM network[51]. Specifically, we generated 1000 surrogate networks by randomly rewiring edges while preserving the nodal degree and approximate edge length distribution of the empirical WM network ("rewired"). The second model evaluated whether the observed correlation was induced by regional correspondence rather than the spatial autocorrelation of CT maturation[45,46]. Specifically, we generated

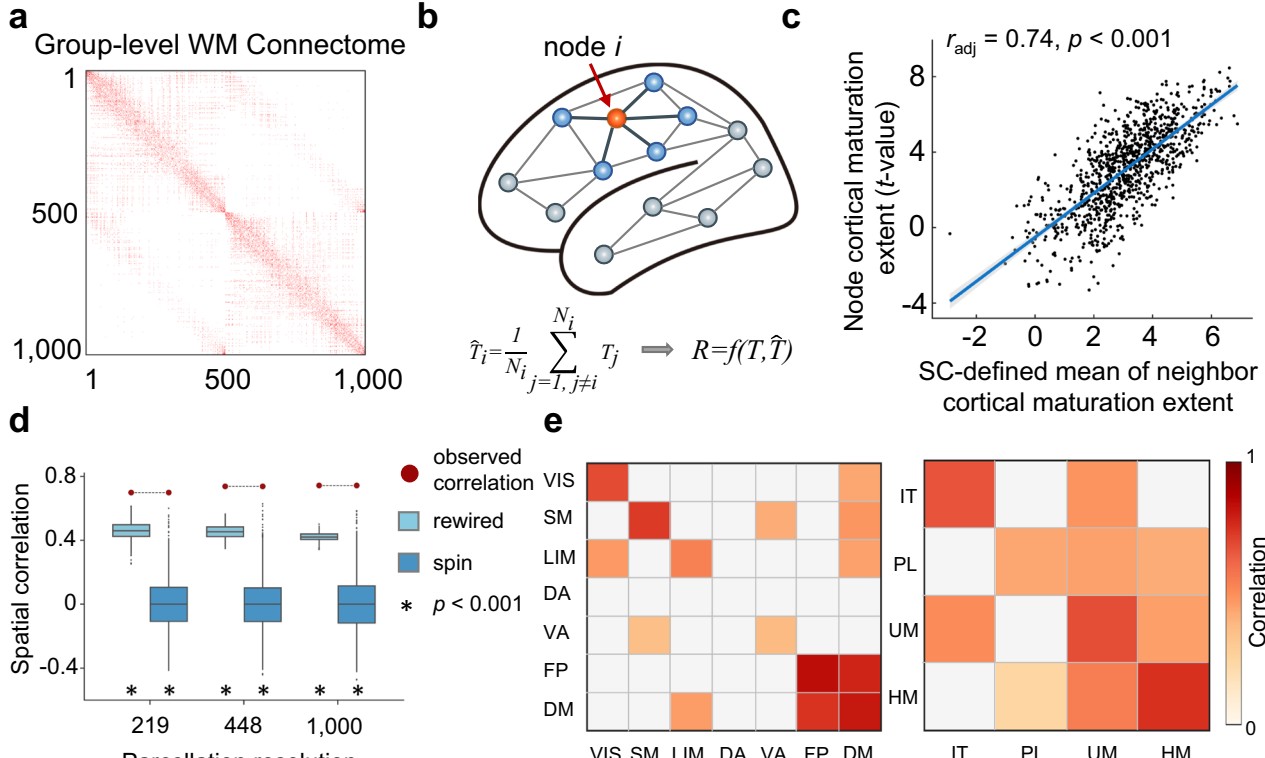

**Fig. 2 | Associations of regional CT maturation with the WM network architecture. a** Group-level connectome backbone at 1000-node resolution. **b** Schematic diagram of WM network-associated CT maturation. The CT maturation extent of a given node (orange) was correlated with the mean maturation extent of its directly connected neighbors (blue) to test whether the maturation of CT was associated with the WM network architecture. **c** A significant correlation was observed between the nodal CT maturation extent (i.e., $t$ value from Statistical Model I) and the mean of its directly connected neighbors (Pearson correlation, $r_{adj} = 0.74$, $p = 5.56 \times 10^{-176}$, two-sided). The scatter plot shows the result at 1000-node resolution (linear fit (central line in blue) with a 95% confidence interval (shadows in gray)). See Fig. S3 for results at other resolutions. **d** The observed Pearson correlations (shown as red circles) were compared against two baseline null models: (1) The "rewired test" by randomly rewiring edges while preserving the nodal degree and edge length distribution of the empirical WM network 1000 times

(light blue boxes); (2) The "spin test" by generating 1000 surrogate maps through randomly rotating region-level cortical $t$ values (deep blue boxes). Boxes represent the interquartile range (IQR), with the median shown as an inside line, while the lower and upper boundaries of the box correspond to the 25th and 75th percentiles. The whiskers extend to the minimum and maximum values within 1.5×IQR, and data points beyond the whiskers are displayed as outliers. Asterisks denote significance level at $p < 0.001$, one-sided. **e** The spatial correlation at the system level against the "spin test" null model. The whole-brain cortical nodes were classified into seven classic communities[44] (left) and four laminar differentiation levels[47] (right), and the statistically significant correlations are shown in light to dark red color (FDR corrected at $p_{spin} = 0.019$, one-sided). VIS, visual; SM, somatomotor; LIM, limbic; DA, dorsal attention; VA, ventral attention; FP, frontoparietal; DM, default mode; IT, idiotypic; PL, paralimbic; UM, unimodal and HM, heteromodal.

1000 surrogate maps by rotating region-level cortical $t$ values ("spin test"). After recalculating the correlation coefficient, we found that the observed correlation was significantly higher than the correlations in both null models, and these results were highly consistent for all three nodal resolutions (all $p_{rewired} < 0.001$ and all $p_{spin} < 0.001$, Fig. 2d). Interestingly, when estimating the spatial correlations at the system level, we found that direct WM connections within the heteromodal area, especially within and between the FP and DM networks, showed strong associations with the maturation of CT (Fig. 2e).

Considering that spatially adjacent nodes may intrinsically exhibit similar cortical development trends, we further performed another two confounding analyses to demonstrate that the observed correlation is not determined by the spatial proximity effect. In the first analysis, we excluded all spatially adjoining neighbors and recalculated the mean CT maturation extent of the remaining structurally connected neighbors for each brain region ("excluded"). In the second analysis, we regressed out the effect of nodal mean Euclidean distance to its connected neighbors from the mean CT maturation extent ("regressed"). After re-estimating the empirical correlation coefficient (1000-node: adjusted $r_{excluded} = 0.60$ and adjusted $r_{regressed} = 0.74$), we repeated two null model tests and found highly consistent results at all three nodal resolutions (all $p_{rewired} < 0.001$, $p_{spin} < 0.001$, Supplementary Fig. S3).

To further validate whether the associations between direct WM connections and regional CT maturation exist throughout 6 to 14 years old, we treated age as a continuous variable (Statistical Model II) instead of dividing participants into two groups. We employed a semiparametric GAM[33] that included sex as a covariate and participant as a random effect to fit the maturation curves of nodal CT with age. Significant effects of age on nodal CT were found in the dorsolateral prefrontal regions, lateral temporal regions, and lateral parietal regions (Fig. 3a, $P < 0.05$, Bonferroni corrected). Two representative fitting curves of nodal CT maturation in the prefrontal cortex and inferior parietal cortex are shown in Fig. 3b. Next, we obtained the maturation rates of nodal CT at each age by calculating the first derivative of the age smooth function. The brain maps of CT maturation rates at three representative ages are shown in Fig. 3c. Finally, we calculated the across-node correlation of the rate of CT maturation between a node and its directly connected neighbor nodes and tested this correlation against two baseline null models at each year of age. We found a significant spatial correlation between the nodal CT maturation rate and the mean of its directly connected neighbors across different age points ($r$: ranged from 0.62 to 0.71, all $p_{spin} < 0.001$, all $p_{rewired} < 0.001$, Fig. 3d).

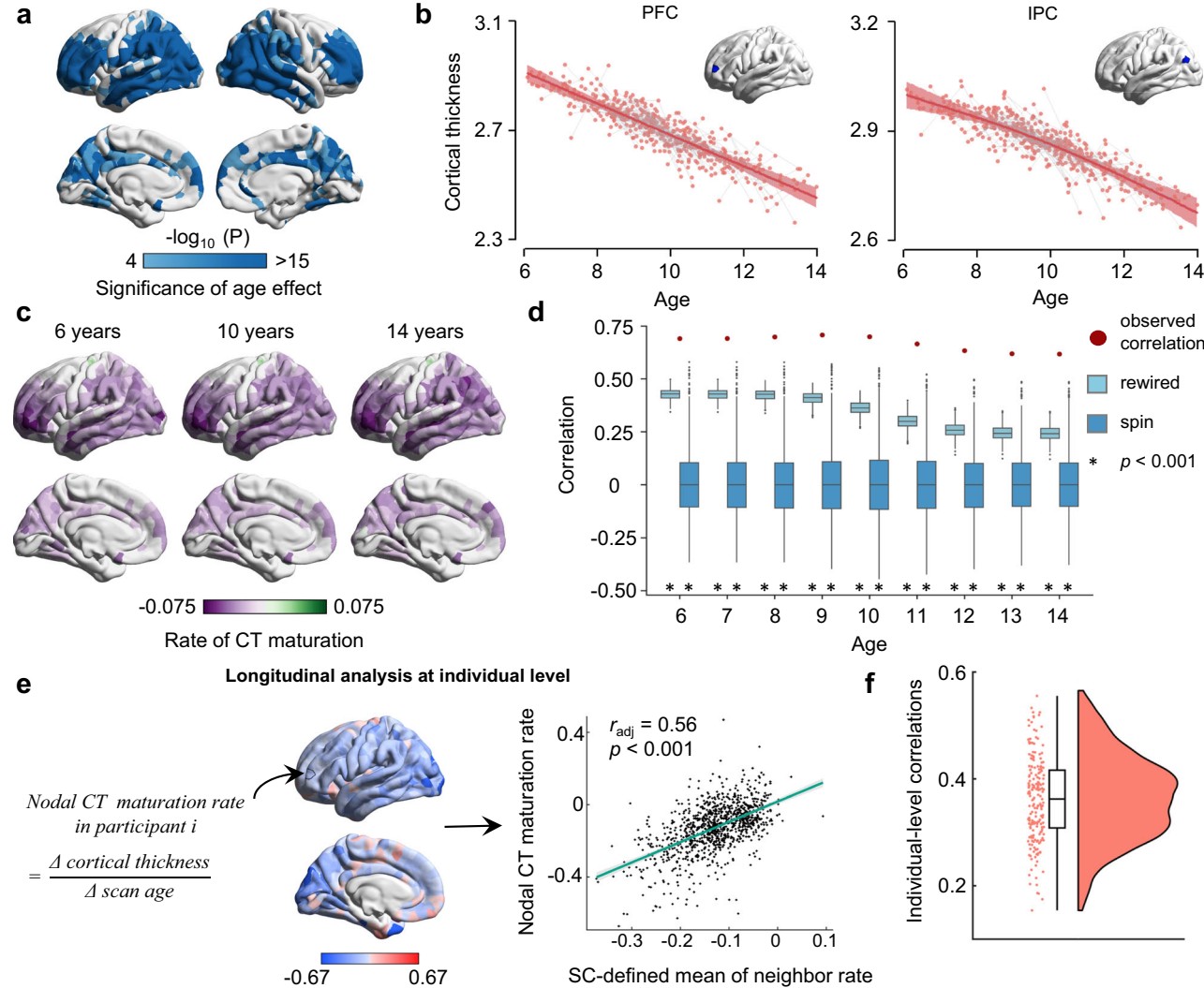

**Fig. 3 | Associations of regional CT maturation with the WM network architecture considering age as a continuous variable or performing individual-level analysis.** (Panel **a** to **d**: Statistical Mode II; Panel **e** and **f**: Statistical Model III). **a** Brain map of regional age effects on CT maturation from childhood to adolescence using the GAM (Bonferroni corrected at $p = 5 \times 10^{-5}$, two-sided). **b** Two representative GAM fitting curves (central lines in deep red) of cortical thinning in the prefrontal cortex (PFC) and inferior parietal cortex (IPC), each with a 95% confidence interval (shadows in light red). **c** Brain maps of regional CT maturation rates (the first derivative of the GAM fitting curves) are shown at representative ages of 6, 10, and 14 years. **d** Significant Pearson correlations were observed between the nodal CT maturation rate and the mean of its directly connected neighbors at each age point. The observed correlations (red dots) were compared to the correlations obtained from 1000 rewired tests (light blue boxes) and 1000 spin tests (deep blue boxes). Asterisks denote statistical significance level at

$p < 0.001$, one-sided. **e** Longitudinal WM network-based CT maturation analysis at the individual level. The definition of nodal CT maturation rates and their brain maps were shown in the left panel. Negative values indicate cortical thinning while positive values indicate cortical thickening with development. The Pearson correlation between nodal CT maturation rate and the mean of its SC-defined neighbors for a representative participant is given in the right panel ($r_{adj} = 0.56$, $p = 3.75 \times 10^{-82}$, two-sided, linear fit (central line in green) with a 95% confidence interval (shadows in gray)). **f** The distribution of these correlations across individuals ($p_{spin} < 0.05$ in all longitudinal samples and $p_{rewired} < 0.05$ in 98.6% (204/207) of longitudinal samples, one-sided). In (**d**, **f**) boxes represent the IQR, with the median shown as an inside line, while the lower and upper boundaries of the box correspond to the 25th and 75th percentiles. The whiskers extend to the minimum and maximum values within 1.5×IQR, and data points beyond the whiskers are displayed as outliers.

Considering the individual differences in cortical maturation[37], we further assessed whether the association between the WM network and CT maturation exists at the individual level by utilizing all individual longitudinal scans independently (105 participants underwent two scans and 51 participants underwent three scans) (Statistical Model III). We first estimated the brain map of nodal CT maturation rates for each individual (Fig. 3e) by calculating the nodal CT difference between two brain scans divided by the gap of scan ages (ΔCT/Δscan age, Fig. 3e left panels). Next, we reconstructed the individual WM networks and repeated the correlation analysis between nodal CT maturation rates and the mean of its directly WM-connected neighbors within each individual (Fig. 3e right panels). We found that this

correlation was significant in almost all individuals (*r*: ranged from 0.15 to 0.56, $p_{spin} < 0.05$ in all longitudinal samples and $p_{rewired} < 0.05$ in 98.6% (204/207) of longitudinal samples, Fig. 3f).

Collectively, these results provide empirical evidence at the network level that the spatial pattern of nodal CT maturation is linked to the WM network architecture.

## The diffusion model of the WM connectome predicts the spatial maturation of CT

To further understand the mechanisms of how the maturation process of cortical morphology is constrained by the WM connectome, we proposed a graph-based diffusion model to simulate the network-level

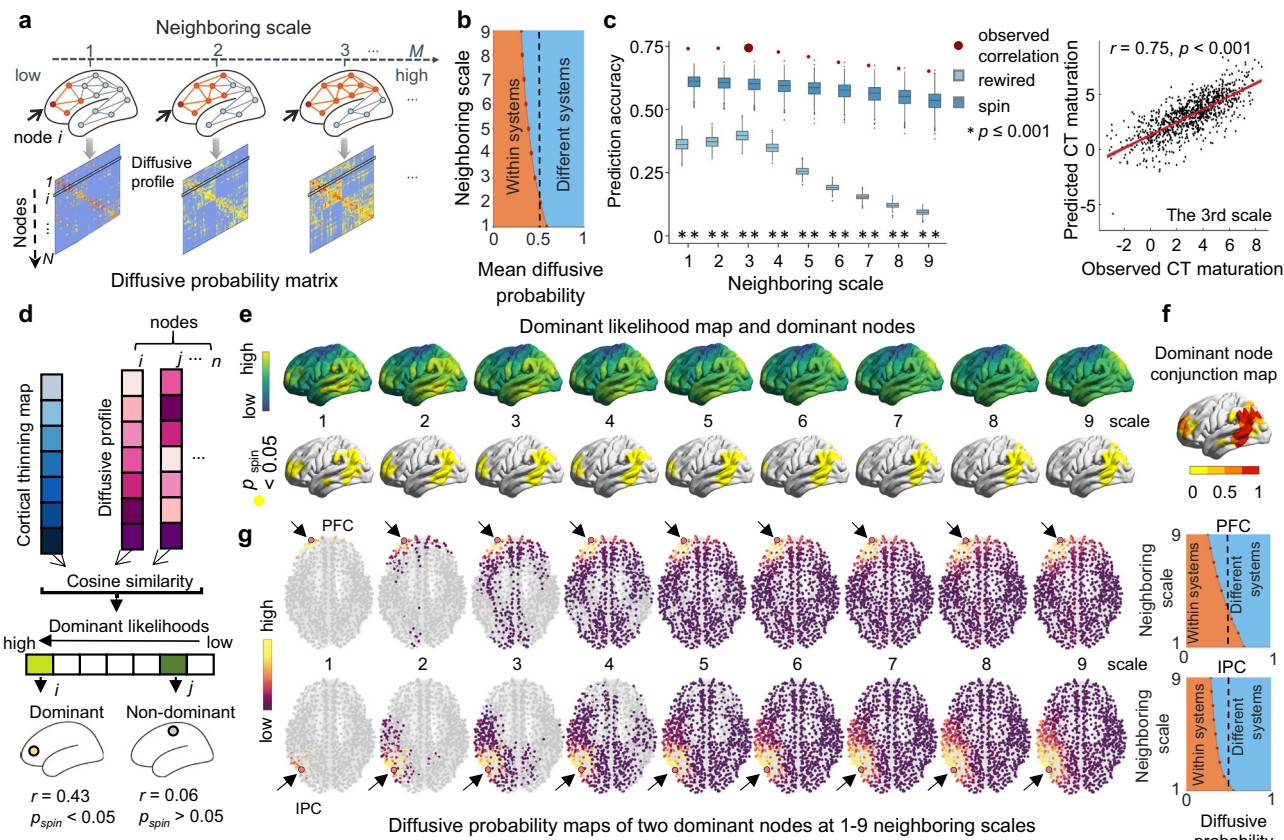

**Fig. 4 | Network-based diffusion model for predicting CT maturation in Statistical Model I. a** Schematic diagram of nodal diffusion processes through multiscale WM edge paths. The orange color represents the edges and nodes at the *mth* neighboring scale of a given node *i* (red). The diffusive profiles of all nodes form the diffusive probability matrix at each neighboring scale. **b** The curve of the average diffusive probability of whole-brain nodes within or between cortical systems. It illustrates that the within-system diffusion probability was greater than 0.5 at the first scale and then decreased along neighboring scales. **c** Significant Pearson correlations between the predicted CT maturation and the observed CT maturation (*t*-value, 1000-node resolution) against 1000 rewired tests (light blue boxes) and 1000 spin tests (deep blue boxes). Boxes represent the IQR with the lower and upper boundaries correspond to the 25th and 75th percentiles, and an inside line indicating the median. The whiskers represent values within 1.5×IQR and individual points represent outliers. Asterisks denote a significance level at $p <= 0.001$, one-sided. The scatter plot depicts the Pearson correlation between actual and predicted CT maturation ($r = 0.75$, $p = 5.11 \times 10^{-178}$, two-sided, linear fit (central line in red) with a 95% confidence interval (shadows in gray)) at the 3rd neighboring scale, which exhibited the highest accuracy. **d** Schematic of the diffusion-based approach used to identify dominant regions. **e** Regional distributions of dominant likelihood (cosine similarity between nodal diffusion profiles and CT maturation map (*t* value)) at neighboring scales of 1−9 (top panels) and the spatial distributions of dominant regions ($p_{spin} < 0.05$, one-sided, bottom panels). **f** The conjunction map of dominant nodes shows the probability of each node being identified as a dominant node across neighboring scales. **g** The diffusive probability maps of two representative dominant nodes separately in the prefrontal (PFC, top panels) and inferior parietal cortex (IPC, bottom panels) at each neighboring scale. Brighter color represents a greater diffusive probability. The right panels show the diffusive probability within or between systems. As the neighborhood scale expands, the diffusion paths of two nodes spreads from local to distributed communities.

axonal interactions during cortical development. The nodal diffusion processes through multiscale WM edge paths were used to predict the maturation of cortical CT. Specifically, we first calculated the diffusive probabilities of a given node to other nodes during a random walk modeling with *m* moving steps (for a toy, see Fig. 4a) to represent the nodal diffusion profile at the *mth* neighboring scale ($m = 1, 2, 3, ...M$; the maximum neighboring scale *M* was set as the network diameter, which is the max shortest path length). Increasing moving steps present expansion scales of the probed neighborhood, which indicates local to distributed preferences of information exchange during the diffusion process. The diffusion profiles of all brain nodes form a diffusive probability matrix that represents the distribution of information propagation throughout the whole network. To further characterize the spatial layout of each diffusive matrix, we classified all cortical nodes into seven brain communities[44] and calculated the average diffusive probabilities within the same system and between different systems separately across brain nodes. Notably, we observed that the diffusion probabilities within the same cortical system were greater than 0.5 at the 1st scale and then decreased with the expansion

of neighboring scales (Fig. 4b). This indicates that a lower scale is mainly involved in more community segregation during propagation. Then, we trained a support vector regression (SVR) model with nodal diffusive profiles at each neighboring scale separately as input features to predict the CT maturation extent from childhood to adolescence (i.e., *t* value from Statistical Model I) in a 10-fold cross-validation strategy[52]. To evaluate the significance of the prediction accuracy, we compared the empirical accuracy with two null model tests, including a spin test and a rewiring test. We found that the diffusive profiles of a given node could significantly predict its CT maturation extent at multiple neighboring scales ($r_{1-9\ scale}$: ranged from 0.65 to 0.75, all $p_{spin} <= 0.001$, all $p_{rewired} < 0.001$, Fig. 4c and Supplementary Table S1). The prediction accuracies were higher at lower neighboring scales. Additionally, features with high contributions to these predictions were mainly involved in the diffusion of frontal and parietal nodes (Supplementary Fig. S4). These results were highly consistent across all three nodal resolutions (Supplementary Fig. S5 and Supplementary Tables S2, S3). Overall, our analysis of computational models indicates that the diffusive characteristics of the WM connectome at the local to

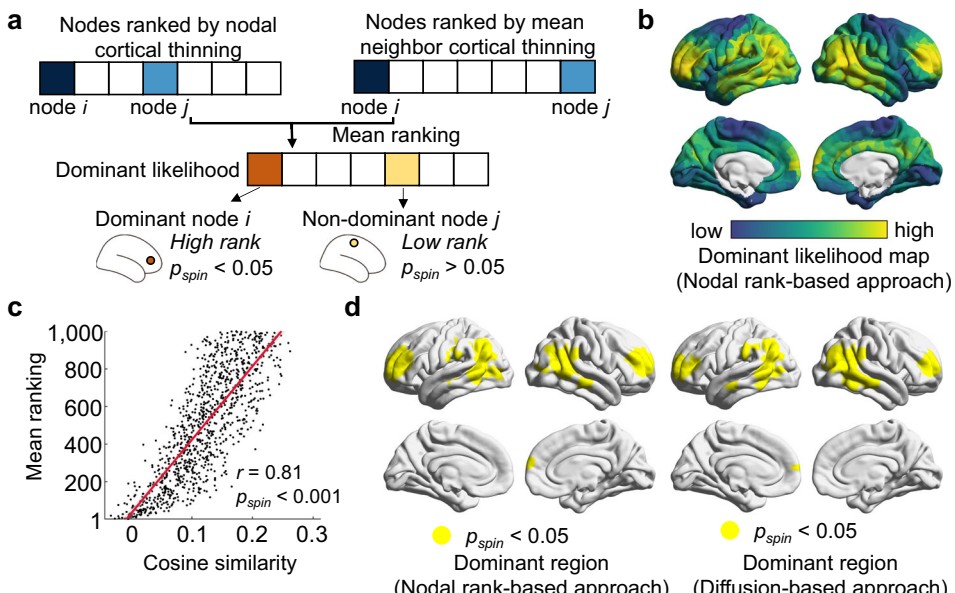

**Fig. 5 | Dominant nodes identified by the nodal rank-based approach.**
**a** Schematic of the method used to identify dominant brain region. A dominant likelihood distribution map was obtained by a nodal rank-based approach introduced by Shafiei et al.,[53] which calculated the mean maturation extents (*t* value from Statistical Model I) in both the focal node and its directly connected neighbors. Regions with significantly higher mean ranks ($p_{spin} < 0.05$) were identified as the dominant nodes. **b** Dominant likelihood distribution map obtained from the nodal rank-based approach. **c** The dominant likelihood maps obtained using the nodal rank-based approach were spatially correlated with the map from our network-based diffusion analysis (Spearman's $r = 0.81$, $p_{spin} < 0.001$ against the null model, one-sided). **d** Dominant region map (left panel, $p_{spin} < 0.05$ against the null model, one-sided) shows a high similarity to the map at the 1st neighboring scale in our network-based diffusion analysis (right panel, $p_{spin} < 0.05$ against the null model, one-sided).

distance scales support the spatial CT maturation map from childhood to adolescence, with a relatively higher effect among nodes within the same cortical system.

We next attempted to measure the dominant likelihood map for the spatial constraint between nodal CT maturation and WM connections and screen out brain nodes that lead the whole brain constraint. For each node, we calculated the cosine similarity between its diffusive profiles at the *mth* (*m* = 1, 2, 3, ..., *M*) scale and the CT maturation extent map from childhood to adolescence (*t*-value in Statistical Model I, Fig. 4d). High similarity of a node indicates that its neighboring diffusion preference largely resembled its neighboring distribution of CT maturation. We observed that the dominant likelihood maps were highly similar across all nine neighboring scales (Fig. 4e, top panel, and Supplementary Fig. S6A) with high values in the bilateral prefrontal, parietal, and temporal regions. These regions were further identified as dominant nodes as they had higher similarity than expected by chance ($p_{spin} < 0.05$) (Fig. 4e, bottom panel, and Supplementary Fig. S6B). The conjunction map of dominant nodes across all neighboring scales is shown in Fig. 4f, where the robust dominant nodes were mainly located in the bilateral prefrontal cortex and inferior parietal cortex. This indicates the leading roles of these regions in shaping the spatial maturation of whole brain CT. Similar results were found in other parcellation resolutions (Supplementary Fig. S7).

To further exemplify the diffusion processes of the dominant nodes at each neighboring scale, we illustrated the diffusive profiles of the two most robust dominant nodes in the prefrontal region (Fig. 4g, top panels) and inferior parietal region (Fig. 4g, bottom panels), respectively. As the neighborhood scale expands, the diffusion of prefrontal dominators mainly spreads to neighbors within FP and DM systems, while the diffusion of parietal dominators mainly spreads to neighbors within DM, DA, and SM systems (Supplementary Fig. S6C). These diffusion processes were mainly involved in nodes within the same system at low neighboring scales and in nodes between systems at high neighboring scales.

nodal rank-based approach. Moreover, to further verify these dominant nodes, we also used a different identification (nodal rank-based) approach[53], which defines dominators as brain regions that show high CT maturation extents in both themselves and their directly connected neighbors (Fig. 5a). Using this approach, we ranked nodes based on their CT maturation extents and their neighbors' mean CT maturation extents separately in ascending order and then calculated the mean rank of each node across both lists. Regions with higher mean ranks ($p_{spin} < 0.05$) were identified as the dominant nodes. We found that this dominant likelihood map was significantly correlated with our network-based diffusion analysis (Spearman's $r = 0.81$, $p_{spin} < 0.001$, Fig. 5b, c), with high consistency of dominant regions (Fig. 5d).

Validating Statistical Model II, we found that nodal diffusive profiles significantly predicted their CT maturation rates (which was obtained from the GAM analysis) at multiple neighboring scales across different ages (all $p_{spin} < 0.01$, $p_{rewired} < 0.001$, Fig. 6a and Supplementary Table S4). Furthermore, we computed the cosine similarity between nodal diffusive profiles at the *mth* (*m* = 1, 2, 3, ..., *M*) scale and the CT maturation rate map, producing the conjunction map of dominant nodes at each age (Fig. 6b). The bilateral prefrontal cortex and inferior parietal regions were consistently identified as major dominant nodes at every age. Thus, our results were robust regardless of whether Statistical Model I or II was used.

### Regional heterogeneous constraints between CT maturation and the connectome are associated with gene expression profiles

Next, we sought to explore the genetic associations of the nodal constraints between the spatial maturation of the CT and WM connectomes during development. We adopted the BrainSpan dataset[54] (Supplementary Section 3.1), which contains gene expression samples of brain tissues from 8 post-conception weeks to 40 years of age, to evaluate the regional genetic relevance. We selected four gene sets following Kang and colleagues[31], which cover typical maturation

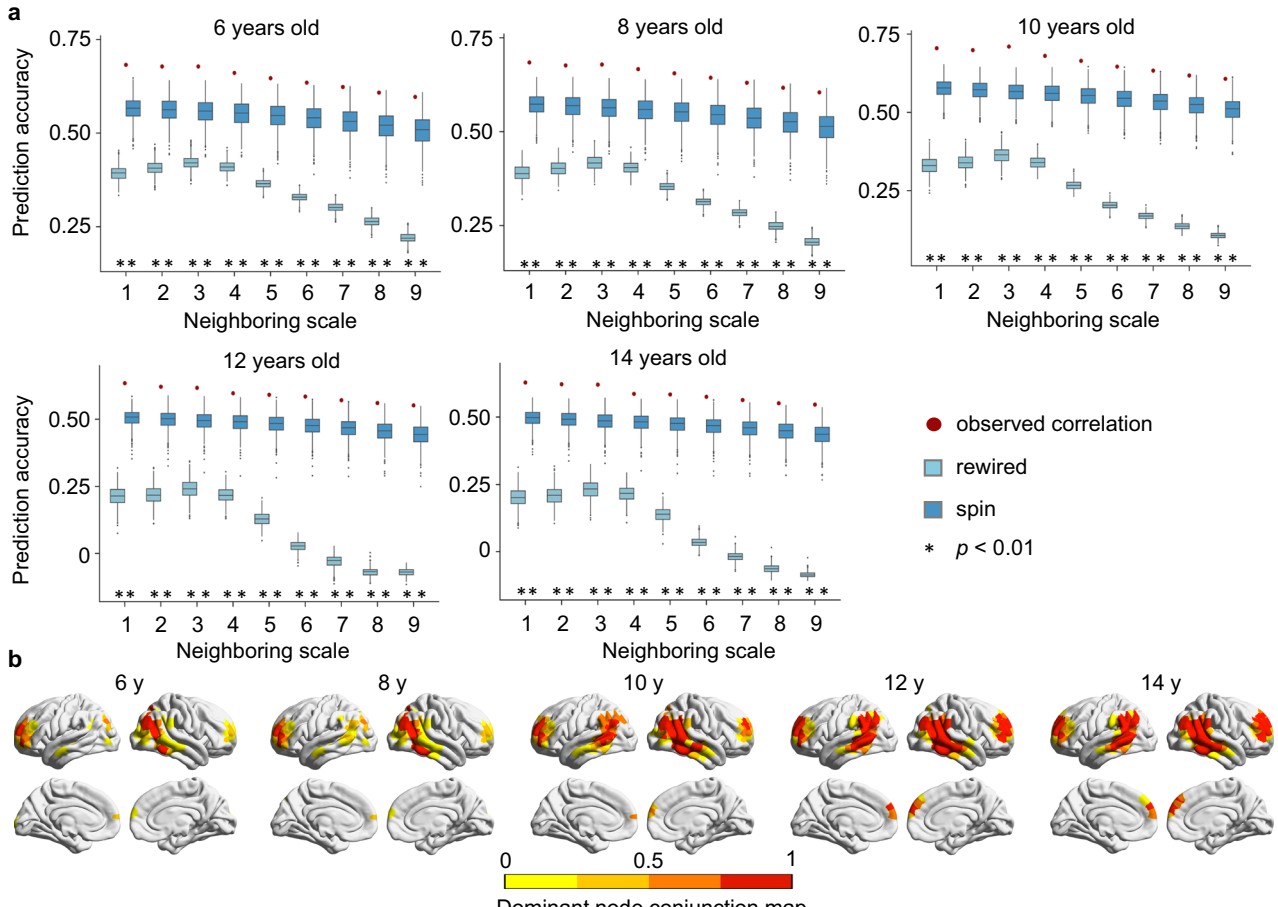

**Fig. 6 | Network-based diffusion model for predicting CT maturation rates in Statistical Model II. a** Significant Pearson correlations between the predicted rate of CT maturation and the observed CT maturation rates (obtained from GAM analysis) at each age. The observed correlations (red dots) were compared to the correlations obtained from 1000 rewired tests (light blue boxes) and 1000 spin tests (deep blue boxes). Boxes represent the IQR, with the median shown as an inside line, while the lower and upper boundaries of the box correspond to the 25th and 75th percentiles. The whiskers extend to the minimum and maximum values within 1.5×IQR, and data points beyond the whiskers are displayed as outliers. Asterisks denote a significance level at $p < 0.01$, one-sided. See Table S4 for detailed $p$ values. **b** The conjunction map of dominant nodes across all nine neighboring scales at each age shows the probability that each node will be identified as a dominant node across scales. y, year.

procedures involved in both CT and WM, including axon development, myelination, dendrite development, and synapse development. We hypothesized that dominant and non-dominant brain nodes should exhibit different transcriptomic characteristics. To this end, we first divided the cortical tissue samples into two categories according to whether the samples were from dominant nodes in the conjunction map (Fig. 4f). Then, we calculated the first principal component score of each gene set's transcription level and estimated the category differences. The statistical significance was calculated by comparing the empirical difference against null differences generated by randomly resampling the same number of genes 1000 times from the remaining genes[9,55,56]. We found divergent transcriptomic trajectories between dominant and non-dominant regions in all four maturation processes from childhood to adolescence (Fig. 7a), and the transcription level in dominant regions was significantly higher than that in nondominant regions for dendrite ($p = 0.014$) and synapse development ($p = 0.002$) but significantly lower for axon development ($p < 0.001$) and myelination ($p < 0.001$) (Fig. 7b). This result indicates that gene expression provides support for the microstructural differences in neurodevelopment between dominant and non-dominant regions, potentially contributing to the non-uniform degree of constraints between CT maturation and the WM pathways.

Considering that the BrainSpan dataset only contains 11 sampling neocortex areas, we also validated the regional gene expression relevance by using Allen Human Brain Atlas (AHBA) datasets[57] Supplementary, Section 3.2. After preprocessing with the abagen (0.1.3) toolbox[58,59], a matrix of gene expression profiles was generated (111 left brain regions × 8631 gene expression levels). Then, we identified the association between the dominant likelihood map and each gene expression map using Pearson's correlation and spin tests (1000 times). A total of 457 genes showed a positive correlation, and 619 genes showed a negative correlation ($p_{spin} < 0.05$, FDR corrected, Supplementary Table S5). Next, we performed Gene Ontology enrichment analysis (Supplementary, Section 3.3) on these two gene sets using the online tools of ToppGene Suite (https://toppgene.cchmc.org/)[60] and REViGO (http://revigo.irb.hr) to select the most meaningful GO terms. We found a significantly correlated gene list with positive correlations mainly enriched in learning or memory and synapse organization (biological process) as well as glutamatergic synapse, neuron spine, and somatodendritic compartment (cellular component) (all $P < 0.05$, FDR corrected, Fig. 7c, d) and negative correlations enriched in the generation of precursor metabolites and energy process (biological process) and myelin sheath components (cellular component) (all $P < 0.05$, FDR corrected, Supplementary

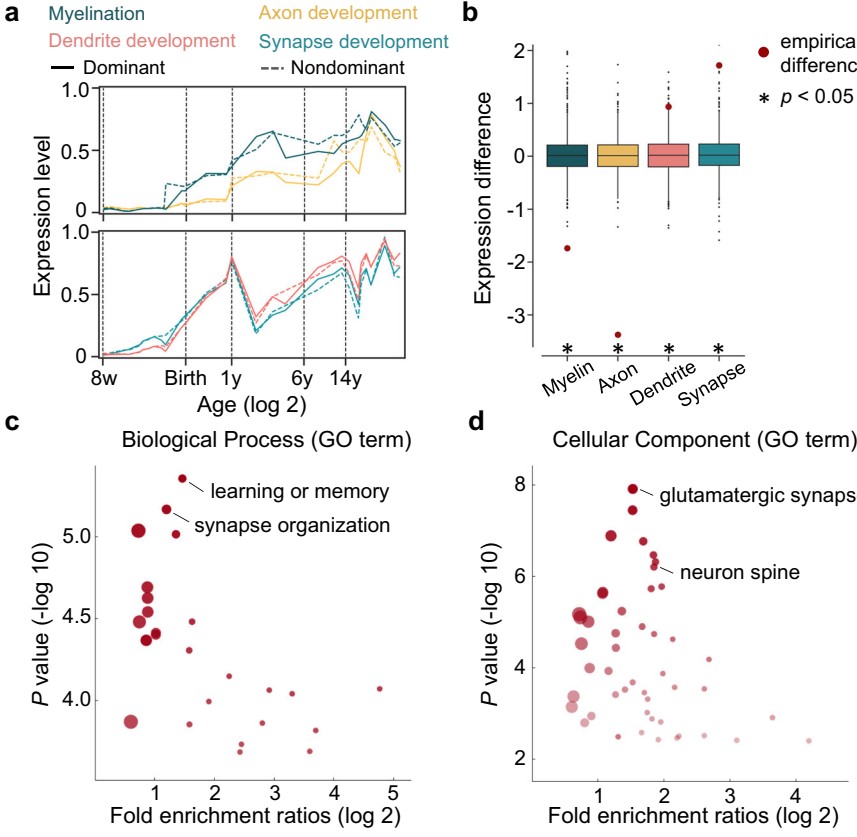

**Fig. 7 | Association between regional heterogeneous constraints and gene expression profiles. a** Transcriptomic trajectories between dominant regions (solid line) and non-dominant regions (dashed line) in four maturation processes. Here, we calculated the first principal component score of each gene set's transcription level. The dominant and non-dominant nodes were determined according to the conjunction map of Fig. 4f. **b** Transcriptomic differences between dominant and non-dominant regions from childhood to adolescence. For each maturation process, the statistical significance was calculated by comparing the empirical difference (red dots) against null differences generated by randomly resampling the same number of genes 1000 times from the remaining genes. In the figure, boxes represent the IQR, with the median shown as a line inside the box, while the lower and upper boundaries of the box correspond to the 25th and 75th percentiles. The whiskers extend to the minimum and maximum values within 1.5×IQR, and individual data points beyond the whiskers are displayed as outliers. Asterisks denote a significance level at $p < 0.05$ ($p$(dendrite) = 0.014, $p$(synapse development) = 0.002, $p$(axon development) <0.001, and $p$(myelination) <0.001, one-sided). **c** Volcano plot depicts Gene Ontology (GO) results for Biological Processes and Cellular Components (**d**). The dots represent the GO terms corrected for multiple comparisons (FDR-corrected at $p < 0.05$, one-sided). The size of the dot indicates the number of genes belonging to the corresponding GO term, and the transparency of the dot represents the significance of the corresponding GO term.

Fig. S8). The detailed enrichment analysis results are shown in Supplementary Tables S6, S7.

## Sensitivity and Replication Analyses

At present, diffusion MRI-based tractography still struggles to accurately reconstruct ultrashort WM fibers[61,62]. To assess the impact of ultrashort streamlines, we first measured the average fiber length between each node and its directly WM-connected neighbors across different nodal parcellations and observed mean values exceeding 40 mm in all three resolutions (mean length: $54.27 \pm 14.35$ mm for 219 nodes, $47.65 \pm 16.65$ mm for 448 nodes, and $41.00 \pm 18.84$ mm for 1000 nodes). This indicates that our network models contained only a few ultrashort streamlines, which may have few impacts on our findings. Additionally, according to recent evidence from superficial WM tracts[63], we excluded streamlines shorter than 20 mm[63] for whole-brain tractography and repeated the diffusion model analysis. We found highly consistent results, indicating that the nodal diffusive profiles still significantly predicted nodal CT maturation extent ($t$ value from Statistical Model I) at multiple neighboring scales ($r_{1-9\ scale}$: ranged from 0.64 to 0.76, all $p_{spin} < 0.001$, all $p_{rewired} < 0.001$, Supplementary Fig. S9).

Considering that the diffusion weighting scheme of the discovery dataset only included b-values of 1000, which could affect the effectiveness of WM tractography[64], we used another independent diffusion imaging dataset with multi-shell diffusion gradients that contain high b-values shells from HCP-D[40] to reconstruct the individual WM network and regenerate the group backbone. The new backbone closely resembled the backbone in our main result. At the global level, the network density of the new backbone (2.38%) was highly similar to that of the backbone in our main results (2.30%). At the nodal and edge levels, the nodal distribution of degree centrality and the edge matrix between two WM backbones both exhibited high similarities (nodal degree: $r = 0.79$, $p_{spin} < 0.001$; edge matrix: $r = 0.75$, $P < 0.001$). Using this new backbone, we found results highly consistent with our main findings. Specifically, nodal CT maturation extents were significantly correlated with their directly connected neighbors (adjusted $r = 0.76$, $p_{spin} < 0.001$, $p_{rewired} < 0.001$, Supplementary Fig. S10A). Using the network-based diffusion model, the spatial maturation of CT was also predicted by the diffusion properties of the WM network ($r_{1-8\ scale}$: ranged from 0.69 to 0.78, all $p_{spin} < 0.001$, $p_{rewired} < 0.001$, Supplementary Fig. S10B and Supplementary Table S8).

To evaluate the reproducibility of our findings, we further replicated all main analyses using the cross-sectional, multi-site replication dataset from HCP-D. The site item was set as a random effect in both linear mixed models (the child group with 98 participants aged

5.58−9.92 y; the adolescent group with 203 participants aged 10.00−14.00 y, Statistical Model I) and GAM models (Statistical Model II) to control for site effects. The results were highly consistent with those obtained using the discovery dataset: (i) several heteromodal areas including dorsolateral prefrontal regions and lateral parietal regions exhibited the most pronounced cortical thinning (Supplementary Fig. S11A); (ii) CT maturation extent ($t$ value) of a node showed a positive correlation with the mean maturation extent of its directly WM connected neighbors (adjusted $r = 0.62$, adjusted $r_{excluded} = 0.46$, and adjusted $r_{regressed} = 0.62$), and the empirical correlation exceeded the values in null models (all $p_{spin} < 0.001$ and all $p_{rewired} < 0.001$, Supplementary Fig. S11B-D); (iii) nodal CT maturation rate was significantly correlated with the mean of its directly WM connected neighbors at each age point ($r$: ranged from 0.41 to 0.66, all $p_{spin} < 0.001$ and all $p_{rewired} < 0.001$, Supplementary Fig. S11E); (iv) the diffusion profiles of the WM network at multiple neighboring scales also predicted the spatial maturation of CT ($r_{1-4\ scale}$: ranged from 0.60 to 0.66, all $p_{spin} < 0.05$, $p_{rewired} < 0.001$, Supplementary Fig. S11F and Supplementary Table S9); and (v) dominant nodes mainly resided in the lateral parietal regions (Supplementary Fig. S11G). Taken together, these findings from an independent replication dataset provide replicable evidence that the WM network structure shapes the cortical maturation from childhood to adolescence.

## Discussion

The present study presents the constraints of the WM network architecture on the coordinated maturation of regional CT from childhood to adolescence and their associations with gene expression signatures. Specifically, we showed that the morphological maturation of cortical nodes is significantly correlated with that of WM-connected neighbors. Moreover, we proposed a network-based diffusion model to predict regional cortical maturation from the WM connectome architecture. Using the WM propagation profiles of brain nodes, this model significantly predicted CT maturation, highlighting the critical role of the WM network architecture in shaping the maturational patterns of cortical morphology. Importantly, these constraints were regionally heterogeneous, with the largest constraints located in frontoparietal nodes, and were associated with the gene expression profiles of microstructural developmental processes. These results were largely consistent across three cortical parcellations and are highly reproducible across independent datasets. Taken together, these findings provide insights into the network-level mechanisms that support the maturation of cortical morphology.

Numerous previous studies have documented that the human brain undergoes remarkable refinements during childhood and adolescence, such as cortical thinning, area expansion, and WM myelination[2,4,7,12,65]. These multifaced gray matter and WM changes have been proven to be intrinsically linked with each other at the regional level. For example, in early childhood, the spatial pattern of cortical surface area expansion during development is highly similar to the myelination of underlying cortico-cortical tracts[66]. In children and adolescents, Jeon et al.[21] reported a significant correlation between the rate of CT decrease and the rate of FA increase in WM tracts at local gyri of the frontal lobe. Ball et al.[67] observed a shared developmental process in CT and structural connectivity during childhood and adolescence. In addition, prior studies show that homologous cortical regions tightly connected by rich WM tracts show high CT maturation couplings[23,25]. At the microscale level, cortical morphology changes during maturation are thought to have various biological origins, including synaptic pruning, increased axon diameter, and myelination[12,68]. Seeking a unified original model for the whole-brain cortical changes is difficult since even within the ventral temporal cortex, thinning of different brain regions seems to be due to distinguished factors[68]. Here, we address this issue with a perspective on brain network modeling. We showed that the morphology maturation

of cortical nodes is well represented by that of their WM-connected neighbors during the transition from childhood to adolescence even after excluding the spatial proximity effect (Fig. 2d Supplementary Fig. S3). It was highly reproducible at different age points, within individuals, and in an independent dataset (Fig. 3d, f, and Supplementary Fig. S11). Such a network-level association is an important extension of previous developmental theories that support cortical thinning across brain development.

The WM network-based cortical maturation could be explained by several factors. First, animal studies revealed that cortical regions that are structurally connected by axon projections are more likely sharing similar cytoarchitectures, such as neuronal density and laminar differentiation[69,70]. Moreover, higher cytoarchitectural similarity among regions tends to higher cortical coordinated maturations[71,72] among neighboring nodes in the brain WM network. Second, a recent study using 19 different neurotransmitter receptors/transporters, such as dopamine and glutamate, found that structurally connected cortical regions usually show greater neurotransmitter receptor similarity[73]. Therefore, these regions may be more inclined to be coregulated by similar physiological processes during development[74,75]. Third, direct WM connections facilitate ongoing interregional communication, enabling these regions to exhibit strong spontaneous neuronal activity couplings[76], which indicates the natural preference for the regional coordination of functional development. This also coincides with Hebbian learning rule, where neurons that fire synchronously tend to form or consolidate connections between them[77,78]. Additionally, WM network-based constraints on cortical morphology exist extensively in adult brains. For instance, Gong et al. suggested that approximately 40% of edges in the adult CT covariance network show matched WM connections[79]. This finding also reflects the close relationship between cortical morphology and the WM network. Such covariation between cortical regions in the adult brain is thought to be associated with their mutual trophic support by axonal pathways[80]. However, whether such cortical covariations originate from the shared constraints of WM connections during development is still an open question. Future studies that combine cortical covariation network models and developmental WM connectomes could help address this important question. In neurodegenerative diseases, including schizophrenia, dementia, and Parkinson's disease, studies have also found that the disease-related cortical deformation pattern across brain regions is conditioned by the WM network[53,81,82].

Notably, in this study, we proposed a network-based diffusion model to explore the constraint of WM on CT maturation. We highlight that nodal diffusion profiles of the WM connectome could accurately predict the maturation pattern of regional CT (Figs. 4c, 6a). From a physical transport perspective, the axonal microenvironment can be regarded as a porous medium that makes diffusion processes within brain tissues extremely critical for delivering oxygen and glucose during neuron metabolism[83]. Meanwhile, diffusion of chemical neurotransmitters at synaptic clefts along axons is essential for forming postsynaptic responses during intercellular communications[84]. At the macroscopic scale, network-based models have been proposed to simulate the consequences of interregional diffusive spread in latent topological space throughout the brain connectome. In neurodegenerative diseases (e.g., Alzheimer's disease), these models showed excellent prediction abilities for the spatial atrophy pattern of the cortex by capturing disrupted transport of trophic factors or accumulated spread of toxic misfolded proteins[85,86]. Based on brain images of nine very prematurely born infants, Friedrichs-Maeder et al. employed a diffusion model to explore the relationship between WM connectivities and cerebral MR measurements such as T1 relaxation time[87]. They reported that early maturation in the primary sensory cortex serves as a source to gradually propagate into the higher-order cortex. In our study, considering the intricate biological relevance between brain WM and cortical morphology[12,68], we used a simple

random walk model to depict the complex network diffusive processes of brain nodes. This model can concisely present the local to distributed supports of the structural connectome on cortical maturation from childhood to adolescence. These nodal diffusive features are effectively integrated by a multivariable machine learning model to represent nodal cortical maturation. Notably, this model showed the significance of indirectly WM-connected neighbors for constraining nodal morphology maturation, which emphasizes the necessity of employing a network-level model to capture this relationship. The contribution from indirect neighboring scales is reasonable because cortical communications between brain regions inherently contain high-order components to support information exchanges between topologically distant nodes[88,89]. In addition, our results provide a detailed description of how these nodes interact with other remote brain nodes through higher-order topological connections during cortical development. These indirect WM neighbors were primarily located within nearby cortical communities (Fig. 4b, c) that share common maturation processes to support morphological integration during cortical development[90,91]. To further examine the association between the WM network topology and cortical maturation, we quantified the correlation between commonly used nodal attributes of the WM network and nodal CT maturation (Supplementary Section 2.2). We selected three nodal topological metrics to measure the capacity of information transmission in common communication dynamics[92,93], including nodal efficiency, the nodal mean first passage time, and the nodal participation coefficient. We found a significant negative correlation between the nodal CT maturation extent (Statistical Model I) and the nodal mean first passage time ($r = -0.22$, $P = 8.52 \times 10^{-13}$, $p_{rewired} < 0.001$ and $p_{spin} = 0.009$) and a significant positive correlation between the nodal CT maturation extent (Statistical Model I) and the nodal participation coefficient ($r = 0.21$, $P = 6.52 \times 10^{-12}$, $p_{rewired} < 0.001$ and $p_{spin} < 0.001$). The correlation between the nodal CT maturation extent and nodal efficiency was not significant ($r = 0.12$, $p_{rewired} = 0.738$ and $p_{spin} = 0.174$). These findings indicate that brain nodes with higher WM network integration capabilities tend to exhibit greater cortical thinning during development, establishing the links between the WM network topology and cortical maturation.

Our results also showed that the constraints of the WM network on CT maturation are spatially heterogeneous (Figs. 4e, 6b). Regionally, dominant nodes in the heteromodal area, especially within and between FP and DM networks, show the strongest spatial constraints. Previous neuroimaging studies have revealed that FP and DM networks display dramatic cortical thinning from childhood to adolescence[1,8]. During the same period, brain WM fractional anisotropy and functional connectivity also show prominently increased tendencies within these networks[94,95]. Our results imply that the WM constraints on the cortical maturation of the heteromodal area likely influence the major pattern of whole-brain cortical thinning. Compatible with our findings of connectome-morphology constraints, structure–function association studies have also shown age-related increases in the heteromodal area during youth, which are associated with individual executive performance[96]. These multifaced heteromodal refinements could support the rapid enhancement of high-order cognitive and social capabilities such as working memory and reasoning[95,97].

By employing transcriptome imaging analyses of data from a developmental gene expression dataset, we found that dominant nodes in the heteromodal area show different transcriptional patterns compared with non-dominant brain nodes. Specifically, dominant regions exhibited higher gene expression levels primarily involved in the maturation of gray matter morphology, including synaptic and dendritic development, and lower expression levels of genes associated with WM maturation, including axon and myelin development. This coincides with findings from histological and MRI brain studies that heteromodal regions have higher synaptic density and lighter

myelination than other regions in childhood and adolescence[28,30], which induces prolonged maturation of the higher-order cortex during adolescence to support the optimization and consolidation of synaptic and axonal connectivity[1,28,30,65]. The non-uniform constraints of WM pathways on CT maturation may be associated with the underlying heterochronous sequence of microstructural development. During adolescence, the heteromodal cortex undergoes more synaptic pruning and reorganization of synapses and dendritic spines than the primary cortex to respond to the demands of cognitive development and environmental experiences[11,30]. At the same time, the WM development in the heteromodal cortex is still incomplete compared to primary cortex[28,65,98]. To consolidate learning and memory[99], the underlying WM pathways in heteromodal regions optimize neural impulse conduction through myelination and increased axon diameter[28,100,101]. For instance, the transmission speed of long-range tracts such as the superior longitudinal fasciculus and temporo-parietal aslant tract that link distributed association cortical regions, increases by approximately twofold[102]. These multifaceted alterations in cortical morphology and WM connectivity in dominant nodes play a crucial role in establishing interregional processing and promoting brain-wide coherence of neural activity[101,103,104]. Likewise, we conducted GO enrichment analysis with the AHBA datasets, which is the most complete gene expression dataset available on the human brain to date, and found that the nonuniform degree of constraints was mainly related to biological processes and cellular components involved in learning or memory, synapse organization, glutamatergic synapse, and neuron spine. These gene-related processes are involved in the spatial thinning of CT during childhood and adolescence[28,29]. As the most abundant synapse type in the neocortex, glutamatergic synapses are primarily responsible for the transport of excitatory transmitters, which are crucial for regulating the transmission and processing of information among brain regions[84]. Meanwhile, neuron spines on dendrites serve to receive various kinds of excitatory inputs from axons and are considered crucial for brain circuit wiring distribution and circuit plasticity[105,106]. Disruptions of these synaptic structures are important substrates of pathogenesis in multiple neurodevelopmental diseases, especially those with deficits in information processing, such as autism[105,107]. There is also another recently emerged approach to identifying genetic influences on brain structure by integrating multi-center brain MR images with genome-wide data from tens of thousands of individuals. Researchers have identified common genetic variations and biological pathways that affect cortical morphology and WM architecture. Interestingly, Grasby et al. found that positive phenotypic correlations were generally observed between spatially adjacent brain areas in terms of regional CT, which also indicates a physical constraint of the genetic influences of cortical morphology[108]. Brouwer et al. utilized longitudinal images and genotyping data covering the lifespan, revealing that the change rate of cortical morphology is under genetic regulation, and such gene associations exhibit age-specific effects[39]. These studies highlight the potential for future utilization of large-sample multimodal brain developmental images combined with genome-wide data to provide deeper insights into the genetic mechanisms underlying the interplay between brain gray matter and WM.

Several issues merit further consideration. First, although dMRI-based deterministic tractography used here is currently a common approach for reconstructing WM tracts in vivo, it is still inherently limited, especially for depicting cross-fibers[109,110]. Although probabilistic tractography approaches exhibit high sensitivity for this issue[111], they result in excessive false positive connections[110]. Second, diffusive processes during axonal transport are proven directional[17]. However, in vivo inference for the direction of WM fibers is still extremely difficult with tractography-based methods. Future investigations combining diffusion models with animal connectome by molecular tracers would reveal a directed network constraint mechanism. Third, dMRI is

an indirect way of assessing the WM microstructure. There still exist many limitations in characterizing intra-axonal properties, particularly at lower diffusion weights, and the interpretation of development-related changes in specific metrics to a precise microstructure event is also difficult[62,112]. Additionally, there is an ongoing debate regarding the appropriate metric for accurately assessing WM connectivity strength in vivo[62,113]. Thus, in the present study, we employed binary networks to capture the backbone of the WM connectome. This approach also simplifies the GAM analysis in part because there was no need to consider the age-dependent interaction of edge strength and constraint degree. In the future, advanced quantitative MRI approaches, such as synthetic MRI, magnetization transfer imaging, and multiexponential T2 imaging, could be utilized to better capture the microstructural properties of brain tissues and further understand the relationship between WM network development and cortical morphological maturation. Fourth, the developmental gene data from BrainSpan only covered 11 areas of the neocortex[54] and thus can only provide a rough exploration of differences in gene expression between dominant and non-dominant nodes. We further validated this result using the AHBA datasets[57], but it was sampled from only six postmortem adult brains. Future studies with gene expression data from widespread cortical regions in a large sample of children and adolescents are important for connectome-transcriptome association analysis. Fifth, due to the mixed design used to collect our data, the developmental effects estimated from the group-level analysis may differ from those observed using pure longitudinal data[37,114,115]. While we validated our findings using longitudinal data from repeated scans within the same participants and obtained consistent results, our findings are limited by the relatively short follow-up periods. In the future, the utilization of longitudinal data with multiple time points and larger sample sizes will be crucial for accurately characterizing individual-level developmental patterns from childhood to adolescence. In addition, our current findings did not allow inference of the causal relationship between the development of the WM network and CT maturation. Implementing the presented methodology using longitudinal data with multiple, densely sampled time points and larger cohorts might provide valuable insights to address this crucial question. Furthermore, utilizing normative models to fit growth charts on a larger sample[116] will be highly important for providing a detailed representation of WM network-constrained cortical maturation. Finally, we showed the constraints of the WM network on cortical morphology maturation during typical development. Previous studies have documented both abnormal cortical maturation and WM connectomes in individuals with neurodevelopmental disorders such as autism[117] and attention-deficit/hyperactivity disorder[118]. In the future, it would be desirable to examine how the WM connectome shapes cortical morphology in these atypical populations.

In conclusion, using neuroimaging, connectomics, transcriptomics, and computational modeling, we found that the maturational pattern of cortical morphology from childhood to adolescence is structurally constrained by the large-scale WM connectome architecture and that such constraints are predominantly located in frontoparietal nodes and are linked with the expression of genes associated with microstructural developmental processes. Thus, our results provide mechanistic insights into the maturation of cortical morphology during development.

## Methods
### Participants and data acquisition
We performed analyses in two independent datasets. After quality control, the discovery dataset included a longitudinal cohort of 314 participants (aged 6–14 years, 161 males and 153 females, sex was self-reported) with 521 structural and diffusion MRI scans from the Beijing Cohort in Children Brain Development project[119]. Among all participants, 158 underwent a single scan, 105 underwent two scans with a mean time interval of 1.16 years, and 51 underwent three scans with an average interval of 0.99 years. Structural and high angular resolution diffusion imaging (HARDI) diffusion MR brain images for each subject were scanned at Peking University using a 3 T Siemens Prisma scanner. Informed written consent was obtained from all participants and at least one parent/guardian, consistent with the guidelines of the Ethics Committee of Beijing Normal University at Beijing Normal University, P. R. China. The replication dataset included a cross-sectional cohort of 301 participants (aged 5–14 years, 183 females and 118 males, sex was self-reported) selected from the Lifespan Human Connectome Project in Development (HCP-D)[40]. Participants were recruited across four imaging sites, and details on imaging protocols can be found in[120].

### MRI Data Preprocessing
For the discovery dataset, individual cortical reconstruction was performed using standard longitudinal processing in the FreeSurfer v6.0 image analysis suite. This process starts with routine steps (recon-all) including intensity normalization, nonbrain tissue removal, tissue segmentation, automated cortical reconstruction, and surface parcellation[42,121–124]. Then, longitudinal streams were performed, including the creation of an unbiased surface template (recon-all -base) and regenerating the cortical surface to reduce variabilities across time points (recon-all -long)[125,126]. Notably, to improve the quality of nonbrain tissue removal, we used HD-BET[127] (https://github.com/NeuroAI-HD/HD-BET), an artificial neural network-based tool, to automatically extract brain tissue images that were further used to replace the brainmask.mgz files in FreeSurfer pipelines. Next, we constructed a custom registration template by averaging all available subjects' cortical surfaces. The atlas in the standard *fsaverage* space was registered to the new custom template and then registered to each subject's surface space to be used to obtain regional CT measurements. All images were visually inspected and manually edited and corrected where needed to ensure the correctness of gray matter and WM boundaries and improve the quality of the output. For diffusion MR images, we employed the standard preprocessing processes in the MRtrix 3.0.1 software[128]. Images were denoised, Gibbs ringing artifacts[129] were removed, and eddy current-induced distortions, head movements, signal dropout, and B1 field inhomogeneity were corrected using MRtrix 3.0.1[128,130–133]. Notably, our dataset acquired additional dual-echo field maps for susceptibility-induced EPI distortion correction. Since such a correction approach is not included in the MRtrix, we fed brain images after eddy correction into the FUGUE process for SIEMENS data in FSL 6.0.1 (https://fsl.fmrib.ox.ac.uk/fsl/fslwiki/FUGUE/Guide#Making_Fieldmap_Images_for_FEAT).

For the replication dataset, the T1-weighted data went through the HCP preprocessing pipeline including the PreFreeSurfer, FreeSurfer, and PostFreeSurfer pipelines[134]. We obtained the individual CT in a common 32k_fs_LR space from the publicly available dataset. For diffusion MR images, we employed the same standard preprocessing steps in the MRtrix 3.0.1 software[128] as the discovery datasets. With one exception, the EPI distortion was corrected by employing TOPUP in MRtrix since the HCP-D dataset contained paired phase-encoded field maps.

### Estimation of regional CT and WM networks
Each participant's cortex was parcellated into 1000 regional nodes with approximately equally sized based on the modified Desikan-Kiliany atlas[42,43] and verified at 219-node and 448-node parcellations. The CT of each brain node was estimated by using FreeSurfer v6.0 software. By using the DSI Studio 2018 software, we reconstructed the whole brain anatomical streamlines using native diffusion MR images for each individual by employing generalized q-sampling imaging (GQI)-based deterministic streamline tractography[49,135] with gray-white boundary as seed voxels. Two cortical regions were considered structurally connected if there exists at least one streamline

with two end points located separately in them[136,137]. After obtaining the individual WM network, we further implemented a consensus approach to generate the binary group-level WM connectome[50].

## Analysis of CT maturation from childhood to adolescence

We explored CT maturation using the following three distinct statistical models. (i) To estimate the maturation of CT from childhood to adolescence, we applied a mixed linear analysis with sex included as a covariate and group included as the main effect for each brain node (Statistical Model I). The model was defined as follows:

$$CT_{ij} = \beta_0 + b_i + (\beta_{group} + b_{group,i}) \cdot group_{ij} + \beta_{sex} \cdot sex_i + \varepsilon_{ij} \quad (1)$$

where $CT_{ij}$ is the CT of participant $i$ at the $j$th scan, $\beta_{group}$ represents the fixed group effect of participant $i$, $b_{group,i}$ is the random effect, and $\varepsilon_{ij}$ is the residual. The $t$ statistics from the group term were used to represent the CT maturation extent of brain nodes. Greater positive $t$-values indicated more significant cortical thinning. (ii) To further validate whether the associations between the WM network architecture and CT maturation were present throughout the ages of 6 to 14 years, we considered age as a continuous variable using semiparametric GAMs[33] to flexibly investigate linear and nonlinear relationships between CT and age (Statistical Model II). For each cortical node, the model was defined as follows:

$$CT_{ij} = \beta_0 + \beta_1 \cdot f_1\left(age_{ij}\right) + \beta_2 \cdot f_2(participant) + \beta_3 \cdot sex_i + \varepsilon \quad (2)$$

where the CT of participant $i$ at the $j$th scan as the dependent variable, age as a smooth term, sex as a linear covariate, and participant as a random effect. Thin plate regression splines were used for the smoothing basis and the residual estimates of maximum likelihood (REML) method was used to estimate the smoothing parameter. Next, we calculated the first derivative of the age smooth function (Δ cortical thickness/Δ scan age) to characterize the CT maturation rate. (iii) To assess whether the association of the WM network and CT maturation also exists at the individual level, we estimated the individual-level CT maturation rates using the longitudinal MRI scan data from each participant in the discovery dataset (Statistical Model III). The three repeated scans from a same participant were split into two continuous pair-scan combinations. Therefore, a total of 207 longitudinal samples were included in this analysis. For each brain node, the CT maturation rate was defined as follows:

$$CT\ maturation\ rate = \frac{CT_{i,j+1} - CT_{i,j}}{scan\ age_{i,j+1} - scan\ age_{i,j}} \quad (3)$$

where $CT_{ij}$ is the CT of participant $i$ at the $j$th scan. Negative values indicate cortical thinning while positive values indicate cortical thickening with development.

## Association between CT Maturation and the WM connectome

To test whether regional maturation of CT was associated with its direct WM connections, we first assessed the across-node relationship between the CT maturation extent ($t$-value between child and adolescent groups, Statistical Model I) of a node and its directly connected neighbor nodes by a model as follows:

$$\hat{T}_i = \frac{1}{N_i} \sum_{j\neq i, j=1}^{N_i} T_j \quad (4)$$

In this model, $\hat{T}_i$ represents the estimated CT maturation extent of node $i$ according to its directly connected neighbors. $T_j$ represents the CT maturation extent ($t$ values as mentioned above) of the $j$th neighbor, and $N_i$ is the number of directly connected neighbors of node $i$.

Specifically, we used the group-level binary WM network to define the WM-connected neighbors of each cortical node. Then, we calculated the spatial correlation between the empirical CT maturation extent (nodal $t$-value) and the estimated values ($\hat{T}_i$). The correlation coefficient was used to represent the association extent between the WM edges and the nodal maturation pattern of CT. We used a similar process to quantify the association degree for each year of CT maturation rates obtained through GAMs (Statistical Model II). The GAMs were performed using the mgcv (1.8.35) R package.

To test whether the association of the WM network and CT maturation was present at the individual level, we conducted an analysis utilizing longitudinal data from repeated scans within each participant (Statistical Model III). We first calculated the annual rate of CT change for each individual to characterize their unique cortical maturation patterns (see Analysis of CT Maturation from Childhood to Adolescence). Next, we reconstructed the individual WM network for each participant from their first scan in each pair-scan combination. Finally, as in the group-level analysis, we assessed the across-node relationship between the annual nodal CT maturation rate and the mean of its directly connected neighbors in each individual's WM network by a model as follows:

$$CT\ \widehat{maturation\ rate}_i = \frac{1}{N_i} \sum_{j\neq i, j=1}^{N_i} CT\ maturation\ rate_j \quad (5)$$

Here, $CT\ \widehat{maturation\ rate}_i$ represents the estimated CT maturation rate of node $i$ according to its directly connected neighbors. $CT\ maturation\ rate_j$ represents the CT maturation rate of the $j$th neighbor, and $N_i$ is the number of directly connected neighbors of node $i$. Then, we calculated the spatial correlation between the empirical CT maturation rate and the estimated values. These correlation coefficients were used to represent the strength of the individual-level association between the WM network and CT maturation.

## Null models

We tested the observed spatial correlation against two baseline null models. In the first null model, we used a spatial permutation test ("spin test") to explore whether the observed correlation was specific to the actual CT maturation pattern rather than due to the spatial autocorrelation of CT maturation[45,46]. Specifically, we first recorded the spherical coordinates of centroids for each parcel in the Cammoun atlas[43]. Then, we randomly rotated the parcels while maintaining spatial autocorrelation and reassigned node values to the nearest parcels. This procedure was repeated 1000 times to create surrogate brain maps. The $p$-value was calculated as the fraction of correlations in null models that exceeded (for positive correlations) or were weaker than (for negative correlations) the observed correlation.

In the second null model, we evaluated whether the observed correlation is determined by the empirical WM network topology rather than the basic spatial embedding of the WM network (such as the distribution of node degree and edge length), we used a rewired null model ("rewired")[51]. Specifically, we first divided edges into different bins according to their Euclidean distance. To preserve the degree sequence and approximate edge length distribution of the empirical WM network, edge pairs were randomly swapped within each bin. Finally, 1000 surrogate networks were generated by repeating this procedure. The $p$-value was calculated as the fraction of correlations in null models that exceeded (for positive correlations) or were weaker than (for negative correlations) the observed correlation.

## Network-based diffusion model

We proposed a diffusion model by combining $m$th-order random walk processes with an SVR method to determine whether the diffusion

properties of the WM network could predict the maturation pattern of CT. Specifically, for an adjacency matrix $A$, the probability of node $i$ transferring to its neighbor $j$ during one step is $A_{ij}/d_i$ (modeled by a random walker moving one step along the edges of the WM network), where $d_i$ is the structurally connected neighbor number (node degree) of node $i$. Thus, the transition probabilities of the WM network were represented by the transition matrix P. P was defined as:

$$P = D^{-1}A \qquad (6)$$

Where $D$ is the node degree diagonal matrix. The initial distribution of random walkers is represented in $p_0$, where the diagonal elements are 1 and the other values are equal to 0. Therefore, when these random walkers move $m$ steps ($m = 1, 2, 3, ...$), their distribution can be described as:

$$P(m) = p_0 P^m \qquad (7)$$

The sum of elements in each row of the distribution matrix is 1, reflecting the diffusion preference of each node with its $mth$-order neighborhoods. Finally, we averaged the outgoing and incoming random walker distribution matrix as a symmetrical diffusion connectivity matrix at each step to represent the bidirectional diffusion properties between any two nodes. Each row of this matrix represents the diffusive profile at the $mth$ neighboring scale of each cortical node.

Next, we trained the SVR model with the diffusion profiles of a brain node separately at each neighboring scale as input features to predict the degree of nodal CT maturation. A total of $M$ models (where $M$ is the maximum neighboring scale) were generated to evaluate the predictive ability of each diffusion scale. Each model was trained using a 10-fold cross-validation strategy with a linear kernel. The coefficient of training error, that is, the C parameter, was selected from among 16 values [$2^{-5}, 2^{-4}, ..., 2^9, 2^{10}$] as in a previous study[52]. The Pearson correlation coefficient between the empirical and predicted CT maturation extents was calculated as the prediction accuracy. Two null models were used to evaluate the significance of prediction accuracy (see Null Models). These analyses were performed using the LIBSVM (3.25) toolbox (https://www.csie.ntu.edu.tw/~cjlin/libsvm/).

### Identifying the dominant regions during development

To further identify the dominant regions, which play more important roles in leading cortical development, we calculated the cosine similarity between the CT maturation map and the nodal diffusion profiles at each random walk step. The statistical significance of the spatial similarity for each brain region was assessed by using a spin test (1000 times, see Null Models). Regions with significantly greater spatial similarity than the regional correspondence induced by the spatial autocorrelation of CT maturation ($p_{spin} < 0.05$) were identified as the dominant regions during development. This analysis was conducted at each neighboring scale. To identify robust dominant nodes across neighboring scales, we generated a dominant node conjunction map, which represents the probability of each node being recognized as dominant across all scales.

We further replicated our results using the other method introduced by[53], which aims to find some brain regions that show high maturation extents in both themselves and their directly connected neighbors. To identify such regions, we ranked the nodes' CT maturation extents and their neighbors' mean CT maturation extents in ascending order. For each node, we calculated the mean rank across both lists. Regions with significantly higher ranks ($p_{spin} < 0.05$) were identified as the dominant regions.

### Analysis of the relationship between heterogeneous connectome constraints on cortical maturation and gene expression profiles

We used developmental gene expression data from BrainSpan[54] to evaluate whether there were differences in the expression levels of genes associated with several neural development events between dominant and non-dominant regions. We divided tissue samples into dominant and non-dominant categories according to their anatomical location (from 11 areas in the neocortex) and arranged them in ascending order based on age to explore the temporal characteristics of gene expression. Next, four typical maturation gene sets[31] were selected (covering axon development, myelination, dendrite development, and synapse development) to evaluate whether there were differences in transcription levels between dominant and non-dominant regions. For each gene set, we performed principal component analysis on the gene expression matrix to calculate the first principal component score of each gene set's transcription level in dominant and non-dominant regions. Then, we calculated the difference between the means of the first principal component scores of the two categories of brain regions. The statistical significance of the category differences was estimated as in previous studies[9,55,56]. First, we computed the empirical difference in the transcription level of a target gene set (the mean of the first principal component scores of one gene set across brain regions) between the two categories of brain regions. Then, we randomly sampled a number of genes equal to those in the target gene set 1000 times from the remaining genes in the BrainSpan datasets to generate 1000 surrogate sets. The difference in the transcription level of each surrogate gene set between the two categories of brain regions was calculated to form a null distribution. Finally, we compared the empirical transcription level differences against the null distributions to obtain statistical significance. Notably, the category of brain regions remained unchanged throughout the entire process. To further validate the relationship between spatial heterogeneity constraints and cortical gene expression levels at the whole-brain level, we performed Pearson correlation analysis with AHBA[57] datasets combined with Gene Ontology enrichment analysis. Specifically, we first identified the association between the dominant likelihood map and each gene expression map using Pearson's correlation. To strictly account for the spatial autocorrelation[138], we determined the significance level of the spatial similarity by comparing the empirically observed correlation to a null distribution obtained by 1000 spatial permutation tests ("spin test")[45,46] in which surrogate maps of brain phenotype were generated while maintaining the spatial autocorrelation properties of the original map. Only genes demonstrating significant correlations ($p_{spin} < 0.05$, FDR corrected) were retained for subsequent GO enrichment analysis[60].

Detailed information about participants, image acquisition, data preprocessing, and data analyses are further described in Supplementary Materials.

### Reporting summary

Further information on research design is available in the Nature Portfolio Reporting Summary linked to this article.

## Data availability

For the Discovery Dataset (CBD dataset) and the Replication Dataset (HCP-D), all data required for reproducing our findings have been publicly available, including the individual regional cortical thickness matrices, structural connectivity matrices, the intermediate results during analysis, and the data for visualizing main figures. They are stored in a publicly accessible cloud repository (https://github.com/Xinyuan-Liang/SC-shapes-the-maturation-of-cortical-morphology). For the Discovery Dataset (CBD dataset), the raw neuroimaging data used in this study are available upon request from the corresponding authors. For the Replication Dataset (HCP-D), raw image scans are

publicly available at https://nda.nih.gov/. The BrainSpan Atlas dataset is publicly available at http://brainspan.org/static/download.html. The AHBA dataset is publicly available at https://human.brain-map.org/static/download. Source data are provided with this paper.

## Code availability

Software packages used in this manuscript include MRtrix 3.0.1 (http://www.mrtrix.org/), FSL 6.0.1 (https://fsl.fmrib.ox.ac.uk/fsl/fslwiki), ANTs 2.3.4 (https://github.com/ANTsX/ANTs), HD-BET (https://github.com/NeuroAI-HD/HD-BET, Jun 16, 2021), FreeSurfer v6.0 (https://surfer.nmr.mgh.harvard.edu/), abagen 0.1.3 (https://github.com/rmarkello/abagen), mgcv 1.8.35 package (https://cran.r-project.org/web/packages/mgcv/index.html), LIBSVM (3.25) (https://www.csie.ntu.edu.tw/~cjlin/libsvm/), BrainNet Viewer 1.7 (https://www.nitrc.org/projects/bnv), R 3.6.3 (https://www.r-project.org), Matlab 2018b (https://www.mathworks.com/products/matlab.html), Python 3.7.12 (https://www.python.org), ToppGene Suit (https://toppgene.cchmc.org/) and REViGO (http://revigo.irb.hr) online tools. The codes used in this study[139] are available at https://github.com/Xinyuan-Liang/SC-shapes-the-maturation-of-cortical-morphology.

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

## Acknowledgements

This work was supported by the scientific and technological innovation 2030 - the major project of the Brain Science and Brain-Inspired Intelligence Technology (No. 2021ZD0200500 (Q.D.)), the National Natural Science Foundation of China (Nos. 82021004 (Y.H.), 31830034 (Y.H.), 31521063 (Q.D.), 31221003 (Q.D.), 32130045 (S.Z.Q.), 81971690 (X.H.L.)), the National Key Research and Development Project (No. 2018YFA0701402 (Y.H.)), Changjiang Scholar Professorship Award (No. T2015027 (Y.H.)), the Beijing Brain Initiative of Beijing Municipal Science & Technology Commission (No. Z181100001518003 (S.T.)), the Fundamental Research Funds for the Central Universities (Nos. 2233300002 (T.D.Z.), 2233100018 (T.D.Z.)), the Beijing Natural Science Foundation (No. JQ23033 (M.R.X.)), and the scientific and technological innovation 2030-Major Projects (No. 2022ZD0211500 (M.R.X.)). We thank Dr. Yongbin Wei and Dr. Yunman Xia for the discussion on gene expression data, Dr. Fan Zhang for the discussion on the construction of superficial WM tracts, and Rui Chen and Haibo Zhang for their helps in data collection. We thank the National Center for Protein Sciences at Peking University in Beijing, China, for assistance with MRI data acquisition. We also thank the Allen Institute for Brain Science for providing the gene expression data. Research reported in this publication was supported by the National Institute of Mental Health of the National Institutes of Health under Award Number U01MH109589 and by funds provided by the McDonnell Center for Systems Neuroscience at Washington University in St. Louis. The content is solely the responsibility of the authors and does not necessarily represent the official views of the National Institutes of Health.

## Author contributions

X.Y.L., T.D.Z. and Y.H. designed research; W.W.M., Y.P.W., S.P.T., J.H.G., S.Z.Q., S.T., Q.D. and Y.H. collected the imaging dataset; T.D.Z., L.L.S., X.H.L., T.Y.L., M.R.X., D.N.D., Z.L.Z., Q.L.L., Z.L.X. and Y.H. provided the methodological instruction; X.Y.L. and T.D.Z. performed the data analysis; X.Y.L., T.D.Z., and Y.H. wrote the paper; X.Y.L., L.L.S., T.D.Z., and Y.H. revised the paper.

## Competing interests

The authors declare no competing interests.
