## [Peer Review File · Nature Communications]

Structural connectome architecture shapes the maturation of cortical morphology from childhood to adolescenceREVIEWER COMMENTS

Reviewer #1 (Remarks to the Author):

In the manuscript entitled “Structural connectome architecture shapes the maturation of cortical morphology from childhood to adolescence” by Liang et al, findings are described of changes in cortical thickness in childhood as compared with adolescence; cross-sectional and longitudinal changes with age in cortical thickness; the relationship with the structural brain network; and links with influences from genetic pathways. This topic is highly interesting and relevant, with a focus on the link between cortical thinning and white matter during human brain development and the influences of genes that may be implicated in these important developmental processes. Multiple different methods are being used and these seem sound and relevant and robust (although the comparison of two groups in a continuous age-cohort is difficult to follow). The longitudinal cohort is interesting. However, some of these methods provide partially overlapping information, and it is not clear why (such as the comparison between two groups and the probably better suited analysis by using the GAM approach). Overall, despite being extensively explained throughout the paper, the methods remain rather difficult to follow. Moreover, several of the findings have been reported before, and this is not always clearly explained. Indeed, the claim made in the paper that these results provide strong evidence from a network level with the spatial pattern of nodal CT maturation structurally constrained by the underlying WM network topology seems overstated, based on the findings presented. The paper would greatly improve if it could remain focused on the new findings (the link between WM and longitudinal CT changes and the associations found using transcriptomics), make clear how these are done methodologically and why these steps were chosen; and make clear how their findings fit into earlier work. In more detail see the points below:

Overall, the statements made based on the data that is being presented are overstated. The claim that the results provide strong evidence from a network level that the spatial pattern of CT maturation is structurally constrained by the underlying WP network topology is interesting and tempting. However, it remains insufficiently clear where the results show this, except for the frontal and parietal cortex thinning that is found to be collectively more pronounced in adolescents than in children and thus is correlated and that these areas are

linked through white matter (known for some time?). The gene expression profile analysis adds to a possible link between CT and WM but as interesting as this analysis is, cannot provide strong evidence either based on these analyses.

The data presented is largely cross-sectional, comparing a group of older children/adolescents (10-14 years of age) with a group of younger children (6-9,9 years of age), whereas in the introduction maturation is being introduced, suggesting longitudinal analyses. For instance, the statement regarding cortical thinning (as visualized in Figure 1) seems based on cross-sectional analyses whereas in the left part of that figure shows longitudinal data-acquisition. Is the longitudinal data entered as a repetition? Please make this very clear in the legend. Some aspects are repeated in a generalized additive model (GAM) and including longitudinal data, which is confusing. The GAM analyses may be the best way to analyze the data, making full use of the breath of the cohort data.

It seems that no longitudinal changes in the WM network strength were included in the analyses, whereas these known to occur during maturation, and seem present in the cohort, why not? Please explain and discuss this.

The network-based diffusion model as presented in Figure 4, includes nearest neighbors that are overall largely at short anatomical distance of the primary node, and it is not clear how this link could have been made through streamlines since such existing local connections are notoriously difficult to measure using DTI. It is indeed done based on the anatomical distance in the T1w images only and not including any streamline information?

The analysis of potential genetic pathways using the gene expression profiles is very interesting. How these differences in four chosen gene sets between the dominant and non-dominant regions in these analyses explains the non-uniform degree of constraints between CT maturation and WM pathways remains somewhat unclear and here the discussion helped with the interpretation of these findings.

Regarding the imaging processing procedures, please be clear that overall standard methodology is used, such as FSL and Freesurfer, for the analyses.

The discussion is overall interesting and relevant although seems to be missing some elements. For instance, a discussion of recent findings on genetic variants associated with cortical thickness (Grasby et al, 2021 Science) and of brain changes including including of the cerebral cortex (Brouwer et al, 2022 Nat Neurosci) is missing in the interpretation of genetic influences on cortical thickness and thinning. Also, a discussion of the findings on CT and WM in light of earlier findings using such approaches, including that of Gong et al, 2012 (cited but not discussed), would help in interpreting these findings in light of other work.

Reviewer #2 (Remarks to the Author):

The current study provides a thorough and novel approach to combining multiple modalities of data to understanding the developmental trajectory of cortical thickness during childhood. Three distinct experiments are explored, looking at: 1) the nearest neighbors prediction of a node's thickness, 2) a network diffusion model showing the local network configuration - not just neighbors - can predict CT trajectory, and 3) linking the development of cortical thickness with gene expression from an existing dataset. Overall, each of the main experiments were explained clearly and given sufficient detail to understand the methods. Each analysis was performed with a thorough cross-validation process and the reporting of all tests was done in a very consistent manner. Where relevant, additional replication samples were used to demonstrate the generalizability of the findings. The datasets utilized are openly available and provide high quality, relevant samples for researchers to utilize them.

I have some very small concerns - addressed below - but I do not believe they pose any meaningful hurdle to publishing the study. I think the results provide an interesting new way to explore the development of cortex development utilizing multiple modalities of information. The writing is very clear and easy to understand.

The datasets utilized are appropriate for answering the questions asked. Different validation sets and findings are used to ensure the generalization of results to other samples in some capacity. A few additional descriptive features would be useful to understand any differences between samples, otherwise the only difference is in the presented novel

findings. Knowing how similar these samples may be from an objective point of view would be helpful. There are a lot of methods described, but I think the article is very clear in its presentation of the different analyses and methods that go into them.

Generally, my comments are relatively minor. The most important changes regard the clarification of some methods, some general descriptive features of the data objects at different stages, and a streamlining of the main and supplemental methods to reduce duplicate text.

(pg 11-12): The process of determining statistical significance of the PCA was done in a way that mirrors the spin/rewired tests of the connectomes, but no reference was provided like for the spin/rewired tests. Is this a novel means of determining significance for transcriptomic data or is there an existing reference? As thorough as the other methods reporting is, it stood out without a reference.

(pg 13) When replicating the structural connectome what other features were inspected? Some basic validation of the network properties would better bolster the claims of replication. Given the scope of the reported findings, having some commonly reported metrics to help compare the two samples would be useful instead of relying solely on the novel (and less familiar) metrics currently used. This applies to any of the replication samples - a basic, quantified descriptive summary would be helpful.

Much of the supplemental methods as they currently exist feel redundant with the main text. Either the supplemental text should be fully combined with the methods section or made to be completely unique.

Minor Things:

The opening paragraph is a bit brief. The last sentence has an abrupt transition. A bit more background that links the many different modes of data discussed for a typical progression would be helpful.

The discussion is strong, but it would benefit from a focused concluding paragraph to bring the main points of the paper back together instead of ending abruptly after discussing some limitations.

(pg. 19) The methods for the "Association between CT Maturation and WM Connectome" need to have the individual analysis explained with a bit more detail. 2 sentences was not

enough.

One of the github links in the submission materials was bad - I got a 404. I did eventually find it, but it would be worth double checking that URL links match the text in the documents.

Overall, I think very minor revisions are needed before publication.

Reviewer #1 (Remarks to the Author):

In the manuscript entitled “Structural connectome architecture shapes the maturation of cortical morphology from childhood to adolescence” by Liang et al, findings are described of changes in cortical thickness in childhood as compared with adolescence; cross-sectional and longitudinal changes with age in cortical thickness; the relationship with the structural brain network; and links with influences from genetic pathways. This topic is highly interesting and relevant, with a focus on the link between cortical thinning and white matter during human brain development and the influences of genes that may be implicated in these important developmental processes. Multiple different methods are being used and these seem sound and relevant and robust (although the comparison of two groups in a continuous age-cohort is difficult to follow). The longitudinal cohort is interesting. However, some of these methods provide partially overlapping information, and it is not clear why (such as the comparison between two groups and the probably better suited analysis by using the GAM approach). Overall, despite being extensively explained throughout the paper, the methods remain rather difficult to follow. Moreover, several of the findings have been reported before, and this is not always clearly explained. Indeed, the claim made in the paper that these results provide strong evidence from a network level with the spatial pattern of nodal CT maturation structurally constrained by the underlying WM network topology seems overstated, based on the findings presented. The paper would greatly improve if it could remain focused on the new findings (the link between WM and longitudinal CT changes and the associations found using transcriptomics), make clear how these are done methodologically and why these steps were chosen; and make clear how their findings fit into earlier work. In more detail see the points below:

R: We thank the reviewer for all positive comments and constructive suggestions. We have addressed these comments in detail as follows.

(1) Overall, the statements made based on the data that is being presented are overstated. The claim that the results provide strong evidence from a network level that the spatial pattern of CT maturation is structurally constrained by the underlying WM network topology is interesting and tempting. However, it remains insufficiently clear where the results show this, except for the frontal and parietal cortex thinning that is found to be collectively more pronounced in adolescents than in children and thus is correlated and that these areas are linked through white matter (known for some time?). The gene expression profile analysis adds to a possible link between CT and WM but as interesting as this analysis is, cannot provide strong evidence either based on these analyses.

R: We sincerely thank the reviewer for these comments and for giving us the opportunity to further clarify the novelty of our findings.

(i) Previously, several studies have reported associations between cortical thickness (CT) maturation and white matter (WM). For instance, decreases in focal CT are related to increased microstructural anisotropy in adjacent WM^{1,2,3,4}. Homologous cortical regions, which are rich in WM fiber connections, exhibit higher CT maturation couplings than nonhomologous regions^{3,5}. However, these previous studies were limited in that they examined associations between CT maturation and WM at only the local level. In this work, we emphasize that the maturation of an individual brain area is not only influenced by one specific neighbor but collectively constrained by all WM-connected neighboring nodes. More importantly, using a network-based diffusion

model, we show that nodal CT maturation from childhood to adolescence can be predicted by nodal diffusion processes through multiscale WM paths in the structural connectome. This diffusion-based computational model provides insights into the network-level mechanisms that support the maturation of cortical morphology. Given the comments raised by this reviewer, we would like to further clarify our primary findings as follows.

First, we demonstrated that the CT maturation extent of a node is highly correlated with the mean CT maturation extent of its directly WM-connected neighbors. This provides direct evidence that the intrinsic wiring pattern of the brain WM network is significantly associated with the spatial patterns of cortical thinning. Notably, as in many brain network studies^{6,7,8}, we employed a null network model by randomly rewiring neighboring nodes. This served as an important reference to verify our findings, and we found significantly stronger empirical CT-WM associations than those in the null model, indicating that the inherent structural wiring pattern of the human brain, not a random wiring pattern, can significantly explain the spatial patterns of cortical maturation. These findings underscore the importance of adopting a network perspective when investigating nodal cortical development.

Second, we used a network-based diffusion model to investigate the mechanisms underlying constraints of WM on cortical maturation. By employing random walks on multiscale pathways in the WM backbone, we obtained a typical topological representation of a given node that reflected the local-to-distributed propagation preferences for information transmission. Notably, just by capturing the WM propagation profiles of brain nodes, this model significantly predicted nodal CT maturation, which highlights the crucial role of the WM network topology in shaping the coordinated developmental patterns of cortical morphology. Previous studies have revealed that the underlying the WM network architecture shapes large-scale neural activity patterns in the developing brain^{9,10}. Here, we provide mechanistic insight into how the architecture of the WM network shapes cortical maturation patterns. Furthermore, we identified dominant nodes in terms of WM constraints on whole-brain CT maturation. Indeed, these dominant nodes were primarily located in frontoparietal association regions, which partly overlap with brain areas previously reported to exhibit the most significant cortical changes during childhood and adolescence. We emphasize that the network diffusion model could provide detailed insights into how these critical frontoparietal nodes interact with other remote brain regions through higher-order topological connections.

Third, to obtain more direct evidence of the association between the WM network topology and cortical maturation, we conducted a new analysis to quantify the correlation between nodal graph-theoretical attributes of the WM network and nodal CT maturation. We selected three nodal topological metrics to measure the regional capacity of information transfer in common communication dynamics^{11,12}, including nodal efficiency (Eff), the nodal mean first passage time (MFPT), and the nodal participation coefficient (PC). We found a significant negative correlation between the nodal CT maturation extent and the nodal MFPT ($r = -0.22$, $P = 8.52 \times 10^{-13}$, $p_{rewired} < 0.001$ and $p_{spin} = 0.009$) and a significant positive correlation between the nodal CT maturation extent and the nodal PC ($r = 0.21$, $P = 6.52 \times 10^{-12}$, $p_{rewired} < 0.001$ and $p_{spin} < 0.001$). The correlation between the nodal CT maturation extent and nodal Eff was not significant ($r = 0.12$, $p_{rewired} = 0.738$ and $p_{spin} = 0.174$). These results showed that nodes with higher diffusive efficiencies and higher network integration capabilities in the WM network tend to exhibit greater cortical thinning during development.

To further improve the clarity of our contributions, we added the relevant Discussion to the

manuscript (Page 16 and Page 18) and presented the new results on Page 18 (further details elaborated in subsequent sections of the text).

(ii) Based on the reviewer's suggestions and the above considerations, we have rephrased the manuscript to remove overstated expressions. We believe that the revised manuscript provides a more balanced and cautious interpretation of the results. The detailed changes are as follows.

[Page 16]

“The present study presents the constraints of the WM network architecture on the coordinated maturation of regional CT from childhood to adolescence and their associations with gene expression signatures. *Specifically, we showed that the morphological maturation of cortical nodes is significantly correlated with that of WM-connected neighbors. Moreover, we proposed a network-based diffusion model to predict regional cortical maturation from the WM connectome architecture. Using the WM propagation profiles of brain nodes, this model significantly predicted CT maturation, highlighting the critical role of the WM network architecture in shaping the maturational patterns of cortical morphology. Importantly, these constraints were regionally heterogeneous, with the largest constraints located in frontoparietal nodes, and were associated with the gene expression profiles of microstructural developmental processes. These results were largely consistent across three cortical parcellations and are highly reproducible across independent datasets.*”

[Page 18]

“*In addition, our results provide a detailed description of how these nodes interact with other remote brain nodes through higher-order topological connections during cortical development. These indirect WM neighbors were primarily located within nearby cortical communities (Fig. 4B-C) that share common maturation processes to support morphological integration during cortical development^{13,14}. To further examine the association between the WM network topology and cortical maturation, we quantified the correlation between commonly used nodal attributes of the WM network and nodal CT maturation (Supplementary Section 2.2). We selected three nodal topological metrics to measure the capacity of information transmission in common communication dynamics^{11,12}, including nodal efficiency, the nodal mean first passage time, and the nodal participation coefficient. We found a significant negative correlation between the nodal CT maturation extent (Statistical Model I) and the nodal mean first passage time ($r = -0.22$, $P = 8.52 \times 10^{-13}$, $p_{\text{rewired}} < 0.001$ and $p_{\text{spin}} = 0.009$) and a significant positive correlation between the nodal CT maturation extent (Statistical Model I) and the nodal participation coefficient ($r = 0.21$, $P = 6.52 \times 10^{-12}$, $p_{\text{rewired}} < 0.001$ and $p_{\text{spin}} < 0.001$). The correlation between the nodal CT maturation extent and nodal efficiency was not significant ($r = 0.12$, $p_{\text{rewired}} = 0.738$ and $p_{\text{spin}} = 0.174$). These findings indicate that brain nodes with higher WM network integration capabilities tend to exhibit greater cortical thinning during development, establishing the links between the WM network topology and cortical maturation.*”

[Page 3]

“If the connectome structure shapes regional cortical maturation, it is necessary to further clarify whether this constraint is *associated with* genetic factors.”

[Page 6]

“Spatial Maturation of CT Links with Direct WM Connections

Next, we tested whether the regional maturation of CT was *associated with* WM network structure.”

[Page 7]

“Figure 2. Associations of regional CT maturation with the WM network architecture. (A) Group-level connectome backbone at 1000-node resolution. **(B)** Schematic diagram of WM network-*associated* CT maturation. The CT maturation extent of a given node (orange) was correlated with the mean maturation extent of its directly connected neighbors (blue) to test whether the maturation of CT was *associated with* the WM network architecture.”

[Page 8]

“Interestingly, when estimating the spatial *correlations* at the system level, we found that direct WM connections within the heteromodal area, especially within and between the FP and DM networks, showed strong *associations with* the maturation of CT (Fig. 2E).

...

To further validate whether the *associations between* direct WM connections *and* regional CT maturation exist throughout 6 to 14 years old, we treated age as a continuous variable (Statistical Model II) instead of dividing participants into two groups.

...

*Considering the individual differences in cortical maturation*¹⁵, we further assess whether the *association between the WM network and CT maturation* exists at the individual level.”

[Page 9]

“Collectively, these results *provided empirical evidence* at the network level that the spatial pattern of nodal CT maturation is *linked to* the WM network architecture.

...

Figure 3. Associations of regional CT maturation with the WM network architecture considering age as a continuous variable or performing individual-level analysis.”

[Page 10]

“Overall, our analysis of computational models indicates that the diffusive characteristics of the WM connectome at the local to distance scales *support* the spatial CT maturation map from childhood to adolescence, with a relatively higher effect among nodes within the same cortical system.”

[Page 14]

“This result indicates that gene expression *provides support for* the microstructural differences in neurodevelopment between dominant and non-dominant regions, *potentially contributing to the* non-uniform degree of constraints between CT maturation and the WM pathways.”

[Page 16]

“These constraints were regionally heterogeneous, with the largest constraints located in

frontoparietal nodes, and *were associated with* the gene expression profiles of microstructural developmental processes.”

[Page 19]

“Our results imply that the WM constraints on the cortical maturation of the heteromodal area *likely influence* the major pattern of whole-brain cortical thinning.”

[Page 21]

“In addition, our current findings did not allow inference of the causal relationship between the development of the WM network and CT maturation. Implementing the presented methodology using longitudinal data with multiple, densely sampled time points and larger cohorts might provide valuable insights to address this crucial question.”

(2) The data presented is largely cross-sectional, comparing a group of older children/adolescents (10-14 years of age) with a group of younger children (6-9,9 years of age), whereas in the introduction maturation is being introduced, suggesting longitudinal analyses. For instance, the statement regarding cortical thinning (as visualized in Figure 1) seems based on cross-sectional analyses whereas in the left part of that figure shows longitudinal data-acquisition. Is the longitudinal data entered as a repetition? Please make this very clear in the legend. Some aspects are repeated in a generalized additive model (GAM) and including longitudinal data, which is confusing. The GAM analyses may be the best way to analyze the data, making full use of the breath of the cohort data.

R: We fully agree with the reviewer that longitudinal scanning data are crucial for understanding the maturation of cortical morphology during children and adolescents. Our study had a mixed longitudinal design as it included multiple structural MRI scans of 314 participants aged 6-14 years. Among these participants, 158 underwent a single scan, 105 underwent two scans separated by a mean time interval of 1.16 years, and 51 underwent three scans separated by an average interval of 0.99 years. We attempted to fully use these longitudinal data to assess nodal CT maturation from childhood to adolescence with three separate statistical models, including group-wise comparisons between children and adolescents (containing all longitudinal scans, Statistical Model I), GAMs fitted across age (Statistical Model II), and individual-level longitudinal analysis (Statistical Model III). In the revised manuscript, we have clarified the use of longitudinal data for all three statistical analyses (Page 4 and Page 6, further details elaborated in subsequent sections of the text).

We would like to further explain why we chose three different analysis methods rather than relying solely on the GAM. Cortical refinements in the period from childhood to adolescence involve substantial changes over a long developmental period (spanning years). At present, it is unclear which statistical models are the most appropriate for capturing cortical development. Therefore, we employed three statistical models with distinct pros and cons to explore the typical cortical CT changing patterns. Specifically, in the revised manuscript, we have extended the GAM analysis as suggested by the reviewer, and we further conducted a diffusion model analysis on the GAM fitting results. We found that the diffusive profiles of a given node could significantly predict its CT maturation rate at multiple neighboring scales across different ages (all $p_{spin} < 0.01$, $p_{rewired} < 0.001$, Fig. 6A, Supplementary Table S4). The bilateral prefrontal cortex and inferior parietal regions consistently emerged as major dominant nodes at every age (Fig.

6B). These new results further demonstrate that WM constraints on cortical morphology maturation are present throughout the developmental stage from childhood to adolescence. We added these results to the revised manuscript on Page 13 (further details elaborated in subsequent sections of the text).

To provide our audience with a clear statement regarding the methodology selected and highlight the importance of the longitudinal fitting, we have revised the manuscript according to the reviewer's suggestion. Relevant passages in the Results (Page 4) and Discussion sections (Page 20) are as follows.

[Page 4]

“To investigate the relationship between cortical morphology maturation and the WM connectome from childhood to adolescence, we leveraged structural and diffusion MRI data from a longitudinal MRI dataset (“discovery dataset”) containing 521 brain scans from 314 participants (aged 6-14 years) in the Children School Functions and Brain Development Project in China (Beijing Cohort) (Fig. 1A). Among the participants, 158 underwent a single scan, 105 underwent two scans separated by an average interval of 1.16 years, and 51 underwent three scans separated by an average interval of 0.99 years.

Currently, it is not clear which statistical models best capture cortical development over time. Therefore, in the present study, we employed three distinct statistical models to assess nodal CT maturation from childhood to adolescence, which include group-wise comparisons between children and adolescents with a mixed linear model¹⁶ (Statistical Model I), generalized additive model (GAM) analysis¹⁷ including age as a continuous variable (Statistical Model II), and individual-level longitudinal analysis using repeated brain imaging scans (Statistical Model III). Our comprehensive analyses aim to yield robust conclusions about the maturational pattern of cortical morphology from childhood to adolescence and how the structural connectome architecture shapes cortical maturation. In Statistical Model I, we employed a group-wise comparison to examine the critical transition from childhood to adolescence. In this analysis, we divided all participants into the child group (218 participants, 299 scans, 6.08-9.98 y) and the adolescent group (162 participants, 222 scans, 10.00-13.99 y) using the age of 10 years as a cut-off, according to the criteria from a previous public health investigation¹⁸ and the World Health Organization (WHO)¹⁹. This method has an advantage in that it is less sensitive to the age distribution of individuals and does not require prior models for fitting age-related growth curves of CT; however, it loses some power to capture fine-grained, age-related change trajectories of CT maturation^{20, 21}. In Statistical Model II, we employed a GAM analysis that treated age as a continuous variable to chart the fine-grained cortical maturation patterns. In this way, we obtained age-related change curves of cortical maturation and investigated how these morphological refinements were constrained by the WM network architecture across different ages. However, this method is sensitive to the sample sizes for each age. In Statistical Model III, we leveraged longitudinal structural MRI data from participants who underwent two or three repeated scans to examine the individual differences in cortical maturation. This individual-based analysis focused on capturing pure longitudinal changes within individuals, minimizing intersubject confounds such as lifestyle and genetic factors^{15, 22, 23}.”

[Page 6]

“(B)Statistical map of CT differences between the child and adolescent groups. *Group differences were used to represent the extent of CT maturation in a mixed linear analysis including sex as a covariate and the individual-specific intercept as a random effect (Statistical Model I).*”

[Page 13]

“Validating *Statistical Model II*, we found that nodal diffusive profiles significantly predicted their CT maturation rates (which was obtained from the GAM analysis) at multiple neighboring scales across different ages (all $p_{spin} < 0.01$, $p_{rewired} < 0.001$, Fig. 6A and Supplementary Table S4). Furthermore, we computed the cosine similarity between nodal diffusive profiles at the n th ($n = 1, 2, 3, \dots, N$) scale and the CT maturation rate map, producing the conjunction map of dominant nodes at each age (Fig. 6B). The bilateral prefrontal cortex and inferior parietal regions were consistently identified as major dominant nodes at every age. Thus, our results were robust regardless of whether *Statistical Model I* or *II* was used.”

Figure 6. Network-based diffusion model for predicting CT maturation rates in *Statistical Model II*. (A) Significant correlations between the predicted rate of CT maturation and the observed CT maturation rates (obtained from GAM analysis) at each age. The observed correlations (red dots) were compared to the correlations obtained from 1000 rewired tests (light blue boxes) and 1000 spin tests (deep blue boxes). In the figure, boxes represent the IQR, with the median shown as a line inside the box, while the lower and upper boundaries of the box correspond to the 25th and 75th percentiles. The whiskers extend to the minimum and maximum values within $1.5 \times IQR$, and individual data points beyond the whiskers are displayed as outliers. Asterisks denote statistical significance ($p < 0.01$). (B) The conjunction map of dominant nodes across all nine neighboring scales at each age shows the probability that each node will be identified as a dominant node across scales. y, year.

[Supplementary Information Page 10]

Table S4. Accuracy of the model at predicting the rates of CT maturation at each age by using multiscale diffusion profiles of network links as features.

		Neighboring scale								
		1	2	3	4	5	6	7	8	9
6 y	r	0.68	0.68	0.68	0.66	0.65	0.64	0.62	0.61	0.60
	p_{spin}	<0.001	<0.001	<0.001	<0.001	<0.001	<0.001	0.001	0.003	0.006
	p_{rewired}	all < 0.001								
8 y	r	0.68	0.68	0.68	0.67	0.66	0.64	0.63	0.62	0.60
	p_{spin}	<0.001	<0.001	<0.001	<0.001	<0.001	<0.001	0.001	0.002	0.004
	p_{rewired}	all < 0.001								
10 y	r	0.70	0.70	0.71	0.68	0.66	0.65	0.63	0.62	0.61
	p_{spin}	<0.001	<0.001	<0.001	<0.001	<0.001	<0.001	<0.001	0.001	0.002
	p_{rewired}	all < 0.001								
12 y	r	0.64	0.62	0.62	0.60	0.59	0.59	0.57	0.56	0.55
	p_{spin}	all < 0.001								
	p_{rewired}	all < 0.001								
14 y	r	0.63	0.62	0.62	0.59	0.58	0.58	0.56	0.55	0.55
	p_{spin}	all < 0.001								
	p_{rewired}	all < 0.001								

Note: The table above shows the prediction accuracies and p-values (calculated as the fraction of null values exceeding the observed accuracy in the “rewired” test and “spin” test) yielded by an SVR model with multiscale diffusion profiles of WM network links as features to predict the CT maturation rates at each age (obtained from Statistical Model II, GAM analysis). Maturation rates of nodal CT and diffusion profiles of WM network links were obtained from the CBD dataset.

[Page 20]

“Fifth, due to the mixed design used to collect our data, the developmental effects estimated from the group-level analysis may differ from those observed using pure longitudinal data^{15, 24, 25}. While we validated our findings using longitudinal data from repeated scans within the same participants and obtained consistent results, our findings are limited by the relatively short follow-up periods. In the future, the utilization of longitudinal data with multiple time points and larger sample sizes will be crucial for accurately characterizing individual-level developmental patterns from childhood to adolescence. ... Furthermore, utilizing normative models to fit growth charts on a larger sample²⁶ will be highly important for providing a detailed representation of WM network-constrained cortical maturation.”

(3) It seems that no longitudinal changes in the WM network strength were included in the analyses, whereas these known to occur during maturation, and seem present in the cohort, why not? Please explain and discuss this.

R: In this study, we did not include longitudinal data on the WM connectivity strength in the analysis for several reasons. First, prior studies have shown that the WM connectivity backbone is relatively constant from late childhood to early adolescence (6 to 14 years old)²⁷. Thus, a unified binary matrix could represent the typical WM linking patterns for this period. Second, there is an ongoing debate concerning the appropriate approach for accurately assessing WM connectivity strength *in vivo*^{28, 29}. Various measurements are currently employed in the field,

such as tractography-based metrics (streamline number, length of streamlines), diffusion tensor-based metrics (fractional anisotropy, apparent diffusion coefficient), and advanced diffusion model-based metrics (e.g., apparent fiber density, intracellular volume fraction)²⁸. It is difficult to determine which metric is the most suitable for describing the WM constraints on cortical maturation, and reaching a solid conclusion requires extensive validation analyses. In addition, since these metrics only provide indirect mappings of the underlying arrangement of axons, the interpretation of development-related changes in specific metrics associated with a precise microstructure event is also challenging^{28,30}. Therefore, we employed a binary WM network backbone in all analyses. This choice also simplified the GAM analysis in part because there was no need to consider the age-dependent interaction of edge strength and WM constraint degree.

Nonetheless, we agree that understanding how longitudinal changes in WM network strengths shape cortical morphology maturation is important. We hope that future studies utilizing advanced quantitative MRI approaches, such as synthetic MRI, magnetization transfer imaging, and multiexponential T2 imaging, can provide more accurate measurements of the “strength” of WM edges (such as myelination) in the pediatric population, could contribute to such an investigation.

We have added this content to the Discussion (Page 20) section as follows.

“Third, dMRI is an indirect way of assessing the WM microstructure. There still exist many limitations in characterizing intra-axonal properties, particularly at lower diffusion weights, and the interpretation of development-related changes in specific metrics to a precise microstructure event is also difficult^{28,30}. Additionally, there is an ongoing debate regarding the appropriate metric for accurately assessing WM connectivity strength in vivo^{28,29}. Thus, in the present study, we employed binary networks to capture the backbone of the WM connectome. This approach also simplifies the GAM analysis in part because there was no need to consider the age-dependent interaction of edge strength and constraint degree. In the future, advanced quantitative MRI approaches, such as synthetic MRI, magnetization transfer imaging, and multiexponential T2 imaging, could be utilized to better capture the microstructural properties of brain tissues and further understand the relationship between WM network development and cortical morphological maturation.”

(4) The network-based diffusion model as presented in Figure 4, includes nearest neighbors that are overall largely at short anatomical distance of the primary node, and it is not clear how this link could have been made through streamlines since such existing local connections are notoriously difficult to measure using DTI. It is indeed done based on the anatomical distance in the T1w images only and not including any streamline information?

R: We thank the reviewer for pointing out this issue. First, we would like to clarify that the network edges in the diffusion computational model were obtained from streamlines of dMRI-based tractography, not from Euclidean distances. We agree with the reviewer that current diffusion MRI-based tractography struggles to accurately reconstruct short-range WM fibers^{28,31}. However, this would not significantly affect our results for the following reasons. First, we employed node parcellations at relatively low resolutions (219 to 1000 nodes) rather than vertex-wise nodal resolutions^{32,33}. In brain WM networks with low nodal resolutions, the contribution of many ultrashort fibers could be largely eliminated because they become within-region connections that are not considered network edges³⁴. Second, the utilization of a group-level

WM backbone also eliminated some inaccurate ultrashort connections even if they were included in individual brain networks. Only the connections that exhibited high reproducibility across the whole population were retained³⁵. To validate these points, we measured the average fiber length between each node and its directly WM-connected neighbors across different nodal parcellations and observed mean values exceeding 40 mm in all three resolutions (mean length: 54.27 ± 14.35 mm for 219 nodes, 47.65 ± 16.65 mm for 448 nodes, and 41.00 ± 18.84 mm for 1000 nodes).

In the revised manuscript, we added a validation test to evaluate the influence of ultrashort streamlines. Several dMRI studies on superficial WM tracts showed that selecting streamlines that exceed 20 mm in length could enable highly reproducible and anatomically reasonable reconstructions of short-range fiber bundles³⁶. Therefore, we excluded streamlines shorter than 20 mm for whole-brain tractography and repeated the diffusion model analysis. We found highly consistent results: the diffusive profiles of a given node significantly predicted its CT maturation extent at multiple neighboring scales ($r_{1-9 \text{ scale}}$: ranged from 0.64 to 0.76, all $p_{spin} < 0.001$, all $p_{rewired} < 0.001$, Supplementary Fig. S9).

We also added these validation results to the ‘‘Sensitivity and Replication Analyses’’ section (Page 15) as follows.

‘‘At present, diffusion MRI-based tractography still struggles to accurately reconstruct ultrashort WM fibers^{28,31}. To assess the impact of ultrashort streamlines, we first measured the average fiber length between each node and its directly WM-connected neighbors across different nodal parcellations and observed mean values exceeding 40 mm in all three resolutions (mean length: 54.27 ± 14.35 mm for 219 nodes, 47.65 ± 16.65 mm for 448 nodes, and 41.00 ± 18.84 mm for 1000 nodes). This indicates that our network models contained only a few ultrashort streamlines, which may have few impacts on our findings. Additionally, according to recent evidence from superficial WM tracts³⁶, we excluded streamlines shorter than 20 mm³⁶ for whole-brain tractography and repeated the diffusion model analysis. We found highly consistent results, indicating that the nodal diffusive profiles still significantly predicted nodal CT maturation extent (t-value from Statistical Model I) at multiple neighboring scales ($r_{1-9 \text{ scale}}$: ranged from 0.64 to 0.76, all $p_{spin} < 0.001$, all $p_{rewired} < 0.001$, Supplementary Fig. S9).’’

[Supplementary Information Page 26]

Fig. S9. Validation of the network-based diffusion model after removing ultrashort WM fibers. Significant correlations between the predicted and the observed CT maturation extent (obtained from Statistical Model I) after removing streamlines shorter than 20 mm in length from the network edges. The observed correlations (red dots) were compared to the

correlations obtained from 1000 rewiring tests (light blue boxes) and 1000 spin tests (deep blue boxes). Asterisks denote statistical significance ($p < 0.001$).

(5) The analysis of potential genetic pathways using the gene expression profiles is very interesting. How these differences in four chosen gene sets between the dominant and non-dominant regions in these analyses explains the non-uniform degree of constraints between CT maturation and WM pathways remains somewhat unclear and here the discussion helped with the interpretation of these findings.

R: According to the reviewer's suggestions, we have updated the Discussion section to provide a more comprehensive interpretation of the observed results. We hope these changes provide greater clarity regarding the genetic associations underlying the non-uniform constraints of WM pathways on CT maturation. The new text is as follows (Page 19).

*“Specifically, dominant regions exhibited higher gene expression levels primarily involved in the maturation of gray matter morphology, including synaptic and dendritic development, and lower expression levels of genes associated with WM maturation, including axon and myelin development. This coincides with findings from histological and MRI brain studies that heteromodal regions have higher synaptic density and lighter myelination than other regions in childhood and adolescence^{37, 38}, which induces prolonged maturation of the higher-order cortex during adolescence to support the optimization and consolidation of synaptic and axonal connectivity^{37, 38, 39, 40}. *The non-uniform constraints of WM pathways on CT maturation may be associated with the underlying heterochronous sequence of microstructural development. During adolescence, the heteromodal cortex undergoes more synaptic pruning and reorganization of synapses and dendritic spines than the primary cortex to respond to the demands of cognitive development and environmental experiences^{37, 41}. At the same time, the WM development in the heteromodal cortex is still incomplete compared to primary cortex^{38, 39, 42}. To consolidate learning and memory⁴³, the underlying WM pathways in heteromodal regions optimize neural impulse conduction through myelination and increased axon diameter^{27, 38, 44}. For instance, the transmission speed of long-range tracts such as the superior longitudinal fasciculus and temporo-parietal aslant tract that link distributed association cortical regions, increases by approximately twofold⁴⁵. These multifaceted alterations in cortical morphology and WM connectivity in dominant nodes play a crucial role in establishing interregional processing and promoting brain-wide coherence of neural activity^{27, 46, 47}.”**

(6) Regarding the imaging processing procedures, please be clear that overall standard methodology is used, such as FSL and Freesurfer, for the analyses.

R: In the revised manuscript, we have revised the Methods section to clarify which procedures were routine and which require special concerns (Page 21).

*“For the discovery dataset, *individual cortical reconstruction was performed using standard longitudinal processing in the FreeSurfer v6.0 image analysis suite* (<https://surfer.nmr.mgh.harvard.edu/>). *This process starts with routine steps (recon-all) including intensity normalization, nonbrain tissue removal, tissue segmentation, automated cortical reconstruction, and surface parcellation^{48, 49, 50, 51, 52}. Then, longitudinal streams were performed, including the creation of an unbiased surface template (recon-all -base) and**

regenerating the cortical surface to reduce variabilities across time points (recon-all -long)^{53, 54}. Notably, to improve the quality of nonbrain tissue removal, we used *HD-BET*⁵⁵, an artificial neural network-based tool, to automatically extract brain tissue images that were further used to replace the *brainmask.mgz* files in *FreeSurfer* pipelines. Next, we constructed a custom registration template by averaging all available subjects' cortical surfaces. The atlas in the standard *fsaverage* space was registered to the new custom template and then registered to each subject's surface space to be used to obtain regional CT measurements. All images were visually inspected and manually edited and corrected where needed to ensure the correctness of gray matter and WM boundaries and improve the quality of the output. For diffusion MR images, we employed the standard preprocessing processes in the *MRtrix 3.0.1* software⁵⁶. Images were denoised, Gibbs ringing artifacts⁵⁷ were removed, and eddy current-induced distortions, head movements, signal dropout, and B1 field inhomogeneity were corrected using *MRtrix 3.0.1*^{56, 58, 59, 60, 61}. Notably, our dataset acquired additional dual-echo field maps for susceptibility-induced EPI distortion correction. Since such a correction approach is not included in the *MRtrix*, we fed brain images after eddy correction into the *FUGUE* process for *SIEMENS* data in *FSL* (https://fsl.fmrib.ox.ac.uk/fsl/fslwiki/FUGUE/Guide#Making_Fieldmap_Images_for_FEAT).

For the replication dataset, the T1-weighted data went through the HCP preprocessing pipeline including the *PreFreeSurfer*, *FreeSurfer*, and *PostFreeSurfer* pipelines⁶². We obtained the individual CT in a common *32k_fs_LR* space from the publicly available dataset. For diffusion MR images, we employed the same standard preprocessing processes in the *MRtrix 3.0.1* software⁵⁶ as the discovery datasets. With one exception, the EPI distortion was corrected by employing *TOPUP* in *MRtrix* since the *HCP-D* dataset contained paired phase-encoded field maps.”

(7) The discussion is overall interesting and relevant although seems to be missing some elements. For instance, a discussion of recent findings on genetic variants associated with cortical thickness (Grasby et al, 2021 *Science*) and of brain changes including of the cerebral cortex (Brouwer et al, 2022 *Nat Neurosci*) is missing in the interpretation of genetic influences on cortical thickness and thinning. Also, a discussion of the findings on CT and WM in light of earlier findings using such approaches, including that of Gong et al, 2012 (cited but not discussed), would help in interpreting these findings in light of other work.

R: We thank this reviewer for pointing out the literature, which has been discussed in the revised manuscript.

[Page 20]

“There is also another recently emerged approach to identifying genetic influences on brain structure by integrating multi-center brain MR images with genome-wide data from tens of thousands of individuals. Researchers have identified common genetic variations and biological pathways that affect cortical morphology and WM architecture. Interestingly, Grasby et al. found that positive phenotypic correlations were generally observed between spatially adjacent brain areas in terms of regional CT, which also indicates a physical constraint of the genetic influences of cortical morphology⁶³. Brouwer et al. utilized longitudinal images and genotyping data covering the lifespan, revealing that the change rate of cortical morphology is under genetic regulation, and such gene associations exhibit age-specific effects²³. These studies

highlight the potential for future utilization of large-sample multimodal brain developmental images combined with genome-wide data to provide deeper insights into the genetic mechanisms underlying the interplay between brain gray matter and WM.”

[Page 17]

“Additionally, WM network-based constraints on cortical morphology exist extensively in adult brains. For instance, Gong et al. suggested that approximately 40% of edges in the adult CT covariance network show matched WM connections ⁶⁴. This finding also reflects the close relationship between cortical morphology and the WM network. Such covariation between cortical regions in the adult brain is thought to be associated with their mutual trophic support by axonal pathways ⁶⁵. However, whether such cortical covariations originate from the shared constraints of WM connections during development is still an open question. Future studies that combine cortical covariation network models and developmental WM connectomes could help address this important question.”

Reviewer #2 (Remarks to the Author):

The current study provides a thorough and novel approach to combining multiple modalities of data to understanding the developmental trajectory of cortical thickness during childhood. Three distinct experiments are explored, looking at: 1) the nearest neighbors prediction of a node's thickness, 2) a network diffusion model showing the local network configuration - not just neighbors - can predict CT trajectory, and 3) linking the development of cortical thickness with gene expression from an existing dataset. Overall, each of the main experiments were explained clearly and given sufficient detail to understand the methods. Each analysis was performed with a thorough cross-validation process and the reporting of all tests was done in a very consistent manner. Where relevant, additional replication samples were used to demonstrate the generalizability of the findings. The datasets utilized are openly available and provide high quality, relevant samples for researchers to utilize them.

I have some very small concerns - addressed below - but I do not believe they pose any meaningful hurdle to publishing the study. I think the results provide an interesting new way to explore the development of cortex development utilizing multiple modalities of information. The writing is very clear and easy to understand. The datasets utilized are appropriate for answering the questions asked. Different validation sets and findings are used to ensure the generalization of results to other samples in some capacity. A few additional descriptive features would be useful to understand any differences between samples, otherwise the only difference is in the presented novel findings. Knowing how similar these samples may be from an objective point of view would be helpful. There are a lot of methods described, but I think the article is very clear in its presentation of the different analyses and methods that go into them.

Generally, my comments are relatively minor. The most important changes regard the clarification of some methods, some general descriptive features of the data objects at different stages, and a streamlining of the main and supplemental methods to reduce duplicate text.

R: We thank the reviewer for all the insightful comments and helpful suggestions.

(pg 11-12): The process of determining statistical significance of the PCA was done in a way that mirrors the spin/rewired tests of the connectomes, but no reference was provided like for the spin/rewired tests. Is this a novel means of determining significance for transcriptomic data or is there an existing reference? As thorough as the other methods reporting is, it stood out without a reference.

R: We thank the reviewer for pointing out this issue. We would like to clarify that the determination of significance for transcriptomic difference between dominant and non-dominant brain regions is not a new approach and has been used in several previous gene-brain association studies^{66, 67, 68}. The rationale of this approach is to estimate whether the transcriptomic characteristics of a target gene set differ from those of surrogate sets. In our work, the detailed testing process was as follows. First, the empirical difference in the transcription level of a target gene set between the two categories of brain regions was calculated (here, the target gene set refers to one of the four gene sets, including axon development, myelination, dendrite development, and synapse development; the difference in the transcription level between the two categories of brain regions refers to the difference in the means of the first principal component scores across brain regions between the dominant and non-dominant categories). Then, we randomly sampled a number of genes equal to the four target gene sets 1000 times from the remaining genes in the BrainSpan datasets to generate 1000 surrogate sets. The difference in the transcription level of each surrogate gene set between the two categories of brain regions was calculated to form a null distribution. Finally, we compared the empirical transcription level differences against the null distributions to obtain statistical significance. Notably, the category of brain regions was unchanged during the whole process, and no spin tests were conducted due to the limited number of tissue locations provided by the BrainSpan datasets (only 11 locations for the two categories). We have updated the description of this method in the main text to make it clear and added the relevant references to the Methods (Page 26) and Results sections (Page 14) as follows.

[Page 26]

“The statistical significance of the category differences was estimated as in previous studies^{66, 67, 68}. First, we computed the empirical difference in the transcription level of a target gene set (the mean of the first principal component scores of one gene set across brain regions) between the two categories of brain regions. Then, we randomly sampled a number of genes equal to those in the target gene set 1000 times from the remaining genes in the BrainSpan datasets to generate 1000 surrogate sets. The difference in the transcription level of each surrogate gene set between the two categories of brain regions was calculated to form a null distribution. Finally, we compared the empirical transcription level differences against the null distributions to obtain statistical significance. Notably, the category of brain regions remained unchanged throughout the entire process.”

[Page 14]

“The statistical significance was calculated by comparing the empirical difference against null differences generated by randomly resampling the same number of genes 1000 times from the remaining genes^{66, 67, 68}.”

(pg 13) When replicating the structural connectome what other features were inspected? Some basic validation of the network properties would better bolster the claims of replication. Given the scope of the reported findings, having some commonly reported metrics to help compare the two samples would be useful instead of relying solely on the novel (and less familiar) metrics currently used. This applies to any of the replication samples - a basic, quantified descriptive summary would be helpful.

R: We thank the reviewer for this constructive suggestion. We added several network metrics from the global to local level to support the replication of the structural connectome between the two datasets, including global network density, nodal degree similarity, and network edge similarity. A summary of the reproducibility of these metrics is also provided in the "Sensitivity and replication analysis" section (Page 16) as follows.

“we used another independent diffusion imaging dataset with multi-shell diffusion gradients that contain high b-values shells from HCP-D⁶⁹ to reconstruct the individual WM network and regenerate the group backbone. The new backbone closely resembled the backbone in our main result. At the global level, the network density of the new backbone (2.38%) was highly similar to that of the backbone in our main results (2.30%). At the nodal and edge levels, the nodal distribution of degree centrality and the edge matrix between two WM backbones both exhibited high similarities (nodal degree: $r = 0.79$, $p_{spin} < 0.001$; edge matrix: $r = 0.75$, $P < 0.001$).”

Much of the supplemental methods as they currently exist feel redundant with the main text. Either the supplemental text should be fully combined with the methods section or made to be completely unique.

R: We have removed redundant content from the supplemental methods. The following sections were either deleted or condensed for brevity: Supplementary, Section 1.1 (Participants), Section 1.3 (MRI Data Preprocessing), Section 1.4 (Analysis of CT Maturation from Childhood to Adolescence), Section 2.2 (Association between CT Maturation and WM Connectome Structure), Section 2.3 (Null Models of Spatial Correlations), Section 2.4 (Network-based Diffusion Model of WM Connectome to Predict Nodal Morphology Maturation), and Section 2.5 (Identifying the Dominant Regions during Development). We hope that these changes have largely improved the readability of the supplemental methods.

Minor Things:

The opening paragraph is a bit brief. The last sentence has an abrupt transition. A bit more background that links the many different modes of data discussed for a typical progression would be helpful.

R: Thanks for this important suggestion. We have revised the opening paragraph as follows (Page 3).

*“The transition period from childhood to adolescence is characterized by prominent reorganization of cortical morphology^{40,70}, which provides critical support for cognitive growth^{71,72}. With progress in modern *in vivo* structural brain imaging, researchers have documented widespread spatial refinements of cortical morphology throughout childhood to adolescence^{20,73}. A typical cortical maturation sequence is marked by hierarchical cortical thinning from the primary cortex to the association cortex^{40,74,75} and is thought to be mediated*

by cellular mechanisms, genetic regulation, and biomechanical factors^{66, 76}. *From a multifaceted developmental perspective, anatomical refinements within neuronal layers at local gyri and sulci, such as synaptic pruning and myelination^{41, 77}, as well as the tension forces exerted by white matter (WM) fibers^{78, 79}, could collectively contribute to cortical maturation. Specifically, WM pathways serve as a structural scaffold for interregional communication, playing a crucial role in supporting the intricate interplay among these factors. Understanding how the brain WM scaffold facilitates anatomical refinements can provide new insights into maturational principles of cortical morphology. In the present study, we present a mechanistic approach to model how the maturational pattern of cortical morphology from childhood to adolescence is shaped by WM connectome architecture.*”

The discussion is strong, but it would benefit from a focused concluding paragraph to bring the main points of the paper back together instead of ending abruptly after discussing some limitations.

R: We have added a paragraph to summarize our findings and insights at the end of the revised manuscript (Page 21) as follows:

“In conclusion, using neuroimaging, connectomics, transcriptomics, and computational modeling, we found that the maturational pattern of cortical morphology from childhood to adolescence is structurally constrained by the large-scale WM connectome architecture and that such constraints are predominantly located in frontoparietal nodes and are linked with the expression of genes associated with microstructural developmental processes. Thus, our results provide mechanistic insights into the maturation of cortical morphology during development.”

(pg. 19) The methods for the “Association between CT Maturation and WM Connectome” need to have the individual analysis explained with a bit more detail. 2 sentences were not enough.

R: We have added relevant descriptions to the “Association between CT Maturation and the WM Connectome” subsection of the Methods in the revised manuscript (Page 24) as follows.

“To test whether the association of the WM network and CT maturation was present at the individual level, we conducted an analysis utilizing longitudinal data from repeated scans within each participant (Statistical Model III). We first calculated the annual rate of CT change for each individual to characterize their unique cortical maturation patterns (see Analysis of CT Maturation from Childhood to Adolescence). Next, we reconstructed the individual WM network for each participant from their first scan in each pair-scan combination. Finally, as in the group-level analysis, we assessed the across-node relationship between the annual nodal CT maturation rate and the mean of its directly connected neighbors in each individual's WM network by a model as follows:

$$CT \widehat{maturation} rate_i = \frac{1}{N_i} \sum_{j \neq i, j=1}^{N_i} CT \text{ maturation rate}_j$$

Here, $CT \widehat{maturation} rate_i$ represents the estimated CT maturation rate of

node i according to its directly connected neighbors. CT maturation rate $_j$ represents the CT maturation rate of the j th neighbor, and N_i is the number of directly connected neighbors of node i . Then, we calculated the spatial correlation between the empirical CT maturation rate and the estimated values. These correlation coefficients were used to represent the strength of the individual-level association between the WM network and CT maturation.”

One of the github links in the submission materials was bad - I got a 404. I did eventually find it, but it would be worth double checking that URL links match the text in the documents.

R: We thank the reviewer for pointing out this issue. We have rechecked and tested the following GitHub links in the revised manuscript, and they currently work well.

1. <https://github.com/Xinyuan-Liang/SC-shapes-the-maturation-of-cortical-morphology>
2. https://github.com/Xinyuan-Liang/SC-shapes-the-maturation-of-cortical-morphology/tree/main/data/gene/AHBA_results.xlsx

Overall, I think very minor revisions are needed before publication.

Reference

1. Jeon T, Mishra V, Ouyang M, Chen M, Huang H. Synchronous changes of cortical thickness and corresponding white matter microstructure during brain development accessed by diffusion MRI tractography from parcellated cortex. *Frontiers in Neuroanatomy* **9**, 158 (2015).
2. Tamnes CK, Østby Y, Fjell AM, Westlye LT, Due-Tønnessen P, Walhovd KB. Brain maturation in adolescence and young adulthood: regional age-related changes in cortical thickness and white matter volume and microstructure. *Cerebral cortex* **20**, 534-548 (2010).
3. Moura LM, *et al.* Coordinated brain development: exploring the synchrony between changes in grey and white matter during childhood maturation. *Brain imaging and behavior* **11**, 808-817 (2017).
4. Vandekar SN, *et al.* Topologically dissociable patterns of development of the human cerebral cortex. *Journal of Neuroscience* **35**, 599-609 (2015).
5. Raznahan A, *et al.* Patterns of coordinated anatomical change in human cortical development: a longitudinal neuroimaging study of maturational coupling. *Neuron* **72**, 873-884 (2011).
6. Betzel RF, Bassett DS. Specificity and robustness of long-distance connections in weighted, interareal connectomes. *Proceedings of the National Academy of Sciences* **115**, E4880-E4889 (2018).
7. Shafiei G, *et al.* Spatial patterning of tissue volume loss in schizophrenia reflects brain network architecture. *Biological psychiatry* **87**, 727-735 (2020).
8. Vázquez-Rodríguez B, *et al.* Gradients of structure–function tethering across neocortex. *Proceedings of the National Academy of Sciences* **116**, 21219-21227 (2019).
9. Baum GL, *et al.* Development of structure–function coupling in human brain networks during youth. *Proceedings of the National Academy of Sciences* **117**, 771-778 (2020).
10. Baum GL, *et al.* Modular segregation of structural brain networks supports the development of executive function in youth. *Current Biology* **27**, 1561-1572. e1568 (2017).
11. Hansen JY, *et al.* Local molecular and global connectomic contributions to cross-disorder cortical abnormalities. *Nature communications* **13**, 4682 (2022).
12. Bullmore E, Sporns O. Complex brain networks: graph theoretical analysis of structural and functional systems. *Nature reviews neuroscience* **10**, 186-198 (2009).
13. Zielinski BA, Gennatas ED, Zhou J, Seeley WW. Network-level structural covariance in

- the developing brain. *Proceedings of the National Academy of Sciences* **107**, 18191-18196 (2010).
14. Nadig A, *et al.* Morphological integration of the human brain across adolescence and adulthood. *Proceedings of the National Academy of Sciences* **118**, e2023860118 (2021).
 15. Becht AI, Mills KL. Modeling individual differences in brain development. *Biological Psychiatry* **88**, 63-69 (2020).
 16. Laird NM, Ware JH. Random-effects models for longitudinal data. *Biometrics*, 963-974 (1982).
 17. Wood SN. Stable and efficient multiple smoothing parameter estimation for generalized additive models. *Journal of the American Statistical Association* **99**, 673-686 (2004).
 18. Sawyer SM, Azzopardi PS, Wickremarathne D, Patton GC. The age of adolescence. *The Lancet Child & Adolescent Health* **2**, 223-228 (2018).
 19. Singh JA, Siddiqi M, Parameshwar P, Chandra-Mouli V. World Health Organization guidance on ethical considerations in planning and reviewing research studies on sexual and reproductive health in adolescents. *Journal of Adolescent Health* **64**, 427-429 (2019).
 20. Bethlehem RA, *et al.* Brain charts for the human lifespan. *Nature* **604**, 525-533 (2022).
 21. Rutherford S, *et al.* Charting brain growth and aging at high spatial precision. *elife* **11**, e72904 (2022).
 22. Vidal-Pineiro D, *et al.* Individual variations in ‘brain age’ relate to early-life factors more than to longitudinal brain change. *elife* **10**, e69995 (2021).
 23. Brouwer RM, *et al.* Genetic variants associated with longitudinal changes in brain structure across the lifespan. *Nature neuroscience* **25**, 421-432 (2022).
 24. Di Biase MA, *et al.* Mapping human brain charts cross-sectionally and longitudinally. *Proceedings of the National Academy of Sciences* **120**, e2216798120 (2023).
 25. Kraemer HC, Yesavage JA, Taylor JL, Kupfer D. How can we learn about developmental processes from cross-sectional studies, or can we? *American Journal of Psychiatry* **157**, 163-171 (2000).
 26. Bozek J, Griffanti L, Lau S, Jenkinson M. Normative models for neuroimaging markers: Impact of model selection, sample size and evaluation criteria. *NeuroImage* **268**, 119864 (2023).
 27. Hagmann P, *et al.* White matter maturation reshapes structural connectivity in the late developing human brain. *Proceedings of the National Academy of Sciences* **107**, 19067-

- 19072 (2010).
28. Zhang F, *et al.* Quantitative mapping of the brain's structural connectivity using diffusion MRI tractography: A review. *Neuroimage* **249**, 118870 (2022).
 29. Yeh CH, Jones DK, Liang X, Descoteaux M, Connelly A. Mapping structural connectivity using diffusion MRI: Challenges and opportunities. *Journal of Magnetic Resonance Imaging* **53**, 1666-1682 (2021).
 30. Jbabdi S, Sotiropoulos SN, Haber SN, Van Essen DC, Behrens TE. Measuring macroscopic brain connections in vivo. *Nature neuroscience* **18**, 1546-1555 (2015).
 31. Guevara M, Guevara P, Román C, Mangin J-F. Superficial white matter: A review on the dMRI analysis methods and applications. *Neuroimage* **212**, 116673 (2020).
 32. Tian Y, Yeo BT, Cropley V, Zalesky A. High-resolution connectomic fingerprints: Mapping neural identity and behavior. *NeuroImage* **229**, 117695 (2021).
 33. Besson P, Lopes R, Leclerc X, Derambure P, Tyvaert L. Intra-subject reliability of the high-resolution whole-brain structural connectome. *NeuroImage* **102**, 283-293 (2014).
 34. Zalesky A, *et al.* Whole-brain anatomical networks: does the choice of nodes matter? *Neuroimage* **50**, 970-983 (2010).
 35. Betzel RF, Griffa A, Hagmann P, Mišić B. Distance-dependent consensus thresholds for generating group-representative structural brain networks. *Network neuroscience* **3**, 475-496 (2019).
 36. Guevara M, *et al.* Reproducibility of superficial white matter tracts using diffusion-weighted imaging tractography. *Neuroimage* **147**, 703-725 (2017).
 37. Huttenlocher PR, Dabholkar AS. Regional differences in synaptogenesis in human cerebral cortex. *Journal of comparative Neurology* **387**, 167-178 (1997).
 38. Whitaker KJ, *et al.* Adolescence is associated with genomically patterned consolidation of the hubs of the human brain connectome. *Proceedings of the National Academy of Sciences* **113**, 9105-9110 (2016).
 39. Miller DJ, *et al.* Prolonged myelination in human neocortical evolution. *Proceedings of the National Academy of Sciences* **109**, 16480-16485 (2012).
 40. Shaw P, *et al.* Neurodevelopmental trajectories of the human cerebral cortex. *Journal of neuroscience* **28**, 3586-3594 (2008).
 41. Norbom LB, *et al.* New insights into the dynamic development of the cerebral cortex in childhood and adolescence: Integrating macro-and microstructural MRI findings.

- Progress in Neurobiology* **204**, 102109 (2021).
42. Lebel C, Treit S, Beaulieu C. A review of diffusion MRI of typical white matter development from early childhood to young adulthood. *NMR in Biomedicine* **32**, e3778 (2019).
 43. Xin W, Chan JR. Myelin plasticity: sculpting circuits in learning and memory. *Nature Reviews Neuroscience* **21**, 682-694 (2020).
 44. Paus T. Growth of white matter in the adolescent brain: myelin or axon? *Brain and cognition* **72**, 26-35 (2010).
 45. van Blooijis D, *et al.* Developmental trajectory of transmission speed in the human brain. *Nature Neuroscience* **26**, 537-541 (2023).
 46. Fornari E, Knyazeva MG, Meuli R, Maeder P. Myelination shapes functional activity in the developing brain. *Neuroimage* **38**, 511-518 (2007).
 47. Spear LP. Adolescent neurodevelopment. *Journal of adolescent health* **52**, S7-S13 (2013).
 48. Fischl B, Dale AM. Measuring the thickness of the human cerebral cortex from magnetic resonance images. *Proceedings of the National Academy of Sciences* **97**, 11050-11055 (2000).
 49. Fischl B, Sereno MI, Dale AM. Cortical surface-based analysis: II: inflation, flattening, and a surface-based coordinate system. *Neuroimage* **9**, 195-207 (1999).
 50. Dale AM, Fischl B, Sereno MI. Cortical surface-based analysis: I. Segmentation and surface reconstruction. *Neuroimage* **9**, 179-194 (1999).
 51. Fischl B, *et al.* Automatically parcellating the human cerebral cortex. *Cerebral cortex* **14**, 11-22 (2004).
 52. Desikan RS, *et al.* An automated labeling system for subdividing the human cerebral cortex on MRI scans into gyral based regions of interest. *Neuroimage* **31**, 968-980 (2006).
 53. Reuter M, Schmansky NJ, Rosas HD, Fischl B. Within-subject template estimation for unbiased longitudinal image analysis. *Neuroimage* **61**, 1402-1418 (2012).
 54. Reuter M, Rosas HD, Fischl B. Highly accurate inverse consistent registration: a robust approach. *Neuroimage* **53**, 1181-1196 (2010).
 55. Isensee F, *et al.* Automated brain extraction of multisequence MRI using artificial neural networks. *Hum Brain Mapp* **40**, 4952-4964 (2019).

56. Tournier JD, *et al.* MRtrix3: A fast, flexible and open software framework for medical image processing and visualisation. *Neuroimage* **202**, 116137 (2019).
57. Kellner E, Dhital B, Kiselev VG, Reisert M. Gibbs-ringing artifact removal based on local subvoxel-shifts. *Magnetic resonance in medicine* **76**, 1574-1581 (2016).
58. Andersson JL, Sotiropoulos SN. An integrated approach to correction for off-resonance effects and subject movement in diffusion MR imaging. *Neuroimage* **125**, 1063-1078 (2016).
59. Andersson JL, Graham MS, Drobnjak I, Zhang H, Filippini N, Bastiani M. Towards a comprehensive framework for movement and distortion correction of diffusion MR images: Within volume movement. *Neuroimage* **152**, 450-466 (2017).
60. Andersson JL, Graham MS, Zsoldos E, Sotiropoulos SN. Incorporating outlier detection and replacement into a non-parametric framework for movement and distortion correction of diffusion MR images. *Neuroimage* **141**, 556-572 (2016).
61. Tustison NJ, *et al.* N4ITK: improved N3 bias correction. *IEEE transactions on medical imaging* **29**, 1310-1320 (2010).
62. Glasser MF, *et al.* The minimal preprocessing pipelines for the Human Connectome Project. *Neuroimage* **80**, 105-124 (2013).
63. Grasby KL, *et al.* The genetic architecture of the human cerebral cortex. *Science* **367**, eaay6690 (2020).
64. Gong G, He Y, Chen ZJ, Evans AC. Convergence and divergence of thickness correlations with diffusion connections across the human cerebral cortex. *Neuroimage* **59**, 1239-1248 (2012).
65. Mechelli A, Friston KJ, Frackowiak RS, Price CJ. Structural covariance in the human cortex. *Journal of Neuroscience* **25**, 8303-8310 (2005).
66. Shin J, *et al.* Cell-specific gene-expression profiles and cortical thickness in the human brain. *Cerebral Cortex* **28**, 3267-3277 (2018).
67. Vidal-Pineiro D, *et al.* Cellular correlates of cortical thinning throughout the lifespan. *Scientific reports* **10**, 21803 (2020).
68. Richiardi J, *et al.* Correlated gene expression supports synchronous activity in brain networks. *Science* **348**, 1241-1244 (2015).
69. Somerville LH, *et al.* The Lifespan Human Connectome Project in Development: A large-scale study of brain connectivity development in 5-21 year olds. *Neuroimage* **183**, 456-468 (2018).

70. Amlien IK, *et al.* Organizing Principles of Human Cortical Development--Thickness and Area from 4 to 30 Years: Insights from Comparative Primate Neuroanatomy. *Cereb Cortex* **26**, 257-267 (2016).
71. Casey BJ, Getz S, Galvan A. The adolescent brain. *Developmental Review* **28**, 62-77 (2008).
72. Shaw P, *et al.* Intellectual ability and cortical development in children and adolescents. *Nature* **440**, 676-679 (2006).
73. Foulkes L, Blakemore S-J. Studying individual differences in human adolescent brain development. *Nature neuroscience* **21**, 315-323 (2018).
74. Sowell ER, Thompson PM, Leonard CM, Welcome SE, Kan E, Toga AW. Longitudinal mapping of cortical thickness and brain growth in normal children. *Journal of neuroscience* **24**, 8223-8231 (2004).
75. Frangou S, *et al.* Cortical thickness across the lifespan: Data from 17,075 healthy individuals aged 3–90 years. *Human brain mapping* **43**, 431-451 (2022).
76. Llinares-Benadero C, Borrell V. Deconstructing cortical folding: genetic, cellular and mechanical determinants. *Nature Reviews Neuroscience* **20**, 161-176 (2019).
77. Walhovd KB, Fjell AM, Giedd J, Dale AM, Brown TT. Through thick and thin: a need to reconcile contradictory results on trajectories in human cortical development. *Cerebral Cortex* **27**, bhv301 (2016).
78. Van Essen DC. A 2020 view of tension-based cortical morphogenesis. *Proceedings of the National Academy of Sciences* **117**, 32868-32879 (2020).
79. Garcia KE, Wang X, Kroenke CD. A model of tension-induced fiber growth predicts white matter organization during brain folding. *Nature communications* **12**, 1-13 (2021).

REVIEWER COMMENTS

Reviewer #2 (Remarks to the Author):

All of my concerns were clearly addressed.

Reviewer #3 (Remarks to the Author):

The authors have written a thorough response to Reviewer contents. The manuscript is interesting and innovative, and I congratulate them on some lovely work. I have only a few comments to improve clarity and rigour:

Lines 906 – 915 – The prediction model is a little unclear. As I understand it, the authors are predicting a vector of maturation values for each region of length N , where N is the number of regions. The predictors are the diffusion profiles, but these are encoded in a $N*N*M$ matrix (Fig 4A), with M being the number of diffusion scales. Please clarify the precise information used in the prediction model.

Lines 920 – 922 – “regions with significantly greater spatial similarity...” greater than what? Also, the wording implies that diffusion profiles were obtained at each diffusion scale. Please clarify whether this analysis was repeated at each scale and, if so, how a specific scale was chosen for further investigation.

The Gene Ontology analysis should account for the potential for false-positive findings, as identified recently (Fulcher et al. Nature Comms, 2021). The current inferential tests are likely to be overly optimistic.

Reviewer #3 (Remarks to the Author):

The authors have written a thorough response to Reviewer contents. The manuscript is interesting and innovative, and I congratulate them on some lovely work. I have only a few comments to improve clarity and rigour:

We thank the reviewer for all the positive comments and helpful suggestions.

1. Lines 906 – 915 – The prediction model is a little unclear. As I understand it, the authors are predicting a vector of maturation values for each region of length N , where N is the number of regions. The predictors are the diffusion profiles, but these are encoded in a $N*N*M$ matrix (Fig 4A), with M being the number of diffusion scales. Please clarify the precise information used in the prediction model.

R: We would like to clarify that we separately established the SVR model at each diffusion scale, and therefore had M models in total (where M is the maximum neighboring scale). For the m th model, the diffusion profiles of the n th node (the n th row vector of $N*N$ diffusion matrix at m th diffusion scale) are utilized as features. We have revised the description of the predictive model section to make it clear (lines 906 – 909). The revised text reads:

“Next, we trained the SVR model with the diffusion profiles of a brain node *separately at each neighboring scale* as input features to predict the degree of nodal CT maturation. *A total of M models (where M is the maximum neighboring scale) were generated to evaluate the predictive ability of each diffusion scale.* Each model was trained using a 10-fold cross-validation strategy with a linear kernel.”

2. Lines 920 – 922 – “regions with significantly greater spatial similarity...” greater than what? Also, the wording implies that diffusion profiles were obtained at each diffusion scale. Please clarify whether this analysis was repeated at each scale and, if so, how a specific scale was chosen for further investigation.

R: We thank the reviewer for pointing out this issue. To identify the dominant regions, we calculated the cosine similarity between the CT maturation map and the nodal diffusion profiles at each scale. The “regions with significantly greater spatial similarity” refer to those regions whose cosine similarity is statistically higher than the regional correspondence induced by the spatial autocorrelation of CT maturation. This evaluation is based on a null distribution generated by the spin test, which randomly permutes the regional labels while maintaining spatial autocorrelation 1000 times. This analysis was repeated at each neighboring scale. The corresponding dominant regions at each neighboring scale were shown in Fig. 4E (also in Supplementary Fig. S6B and Supplementary Fig. S7). To further identify a robust representation of dominant nodes across all neighboring scales, we calculated the probability of each node being recognized as a dominant node across all scales and generated a dominant node conjunction map (Fig. 4F) for subsequent analyses. We have rephrased these descriptions in the Method part to make it clear as follows (lines 919 – 925).

“The statistical significance of the spatial similarity for each brain region was assessed by using a spin test (1000 times, see Null Models). Regions with significantly greater spatial similarity than the *regional correspondence induced by the spatial autocorrelation of CT*

maturation ($p_{spin} < 0.05$) were identified as the dominant regions during development. This analysis was conducted at each neighboring scale. To identify robust dominant nodes across neighboring scales, we generated a dominant node conjunction map, which represents the probability of each node being recognized as dominant across all scales.”

3. The Gene Ontology analysis should account for the potential for false-positive findings, as identified recently (Fulcher et al. Nature Comms, 2021). The current inferential tests are likely to be overly optimistic.

R: We thank the reviewer for the valuable comments. In this literature, Fulcher et al. suggested that researchers need to employ spatial embedding null models to account for the potential false-positive findings of standard Gene-category enrichment analysis. In fact, we have controlled the effect of spatial autocorrelation when estimating the spatial correlation between genes maps and brain phenotype map in our GO enrichment analysis. In the original manuscript, this method has been addressed in the Supplemental Materials but was not described in the main text. In the revised manuscript, we have moved the relevant method descriptions from the Supplementary Materials to the Methods section of the main text (lines 954-964).

“To further validate the relationship between spatial heterogeneity constraints and cortical gene expression levels at the whole-brain level, we performed Pearson correlation analysis with Allen Human Brain Atlas datasets¹ combined with Gene Ontology enrichment analysis. Specifically, we first identified the association between the dominant likelihood map and each gene expression map using Pearson’s correlation. To strictly account for the spatial autocorrelation², we determined the significance level of the spatial similarity by comparing the empirically observed correlation to a null distribution obtained by 1000 spatial permutation tests (“spin test”)^{3,4} in which surrogate maps of brain phenotype were generated while maintaining the spatial autocorrelation properties of the original map. Only genes demonstrating significant correlations ($p_{spin} < 0.05$, FDR corrected) were retained for subsequent GO enrichment analysis⁵.”

Reference

1. Hawrylycz MJ, et al. An anatomically comprehensive atlas of the adult human brain transcriptome. *Nature* **489**, 391-399 (2012).
2. Fulcher BD, Fornito A. A transcriptional signature of hub connectivity in the mouse connectome. *Proceedings of the National Academy of Sciences* **113**, 1435-1440 (2016).
3. Vasa F, et al. Adolescent Tuning of Association Cortex in Human Structural Brain Networks. *Cereb Cortex* **28**, 281-294 (2018).
4. Alexander-Bloch AF, et al. On testing for spatial correspondence between maps of human brain structure and function. *Neuroimage* **178**, 540-551 (2018).
5. Chen J, Bardes EE, Aronow BJ, Jegga AG. ToppGene Suite for gene list enrichment analysis and candidate gene prioritization. *Nucleic acids research* **37**, W305-W311

(2009).